# Minian, an open-source miniscope analysis pipeline

**Zhe Dong[1], William Mau[1], Yu Feng[1], Zachary T Pennington[1], Lingxuan Chen[1], Yosif Zaki[1], Kanaka Rajan[1], Tristan Shuman[1], Daniel Aharoni[2]\*, Denise J Cai[1]\***

[1]Nash Family Department of Neuroscience, Icahn School of Medicine at Mount Sinai, New York, United States; [2]Department of Neurology, David Geffen School of Medicine, University of California, Los Angeles, Los Angeles, United States

**Abstract** Miniature microscopes have gained considerable traction for in vivo calcium imaging in freely behaving animals. However, extracting calcium signals from raw videos is a computationally complex problem and remains a bottleneck for many researchers utilizing single-photon in vivo calcium imaging. Despite the existence of many powerful analysis packages designed to detect and extract calcium dynamics, most have either key parameters that are hard-coded or insufficient step-by-step guidance and validations to help the users choose the best parameters. This makes it difficult to know whether the output is reliable and meets the assumptions necessary for proper analysis. Moreover, large memory demand is often a constraint for setting up these pipelines since it limits the choice of hardware to specialized computers. Given these difficulties, there is a need for a low memory demand, user-friendly tool offering interactive visualizations of how altering parameters at each step of the analysis affects data output. Our open-source analysis pipeline, Minian (miniscope analysis), facilitates the transparency and accessibility of single-photon calcium imaging analysis, permitting users with little computational experience to extract the location of cells and their corresponding calcium traces and deconvolved neural activities. Minian contains interactive visualization tools for every step of the analysis, as well as detailed documentation and tips on parameter exploration. Furthermore, Minian has relatively small memory demands and can be run on a laptop, making it available to labs that do not have access to specialized computational hardware. Minian has been validated to reliably and robustly extract calcium events across different brain regions and from different cell types. In practice, Minian provides an open-source calcium imaging analysis pipeline with user-friendly interactive visualizations to explore parameters and validate results.

**\*For correspondence:**
dbaharoni@gmail.com (DA);
denisecai@gmail.com (DJC)

## Editor's evaluation

An increasing number of systems neuroscience experiments involve imaging neural activity using head-mounted epifluorescent microscopes, or "miniscopes". A growing community has been using the Minian software package to process the imaging data into a useful form for subsequent analysis. The Minian team has done an excellent job of exposing the various parameters involved to be easily accessible to users while maintaining a performant robust tool. This work presents Minian and is in many ways an exemplar of how open source software can be presented in journal form.

## Introduction
### Overview of related works

Open-source projects – hardware, software, training curricula – have changed science and enabled significant advances across multiple disciplines. Neuroscience, in particular, has benefited tremendously from the open-source movement. Numerous open-source projects have emerged (**White**

*et al., 2019*; *Freeman, 2015*), including various types of behavioral apparatus facilitating the design of novel experiments (*Buccino et al., 2018*; *Frie and Khokhar, 2019*; *Lopes et al., 2015*; *Nguyen et al., 2016*; *Matikainen-Ankney et al., 2021*), computational tools enabling the analysis of large-scale datasets (*Kabra et al., 2013*; *Mathis et al., 2018*; *van den Boom et al., 2017*; *Zhou et al., 2018*; *Lu et al., 2018*; *Mukamel et al., 2009*; *Jun et al., 2017*; *Klibisz et al., 2017*; *Giovannucci et al., 2019*; *Sheintuch et al., 2017*; *Pennington et al., 2019*; *Friedrich et al., 2017*; *Giovannucci et al., 2017*; *Jewell and Witten, 2018*), and recording devices allowing access to large populations of neurons in the brain (*Aharoni et al., 2019*; *Owen and Kreitzer, 2019*; *Siegle et al., 2017*; *Solari et al., 2018*; *Barbera et al., 2019*; *Jacob et al., 2018*; *Liberti et al., 2017*; *de Groot et al., 2020*; *Skocek et al., 2018*; *Scott et al., 2018*). Miniature microscopy has been an area of particular importance for the open-source movement in neuroscience. To increase the usability, accessibility, and transparency of this remarkable technology originally developed by Schnitzer and colleagues (*Ghosh et al., 2011*; *Ziv et al., 2013*), a number of labs innovated on top of the original versions with open-source versions (*Barbera et al., 2019*; *Jacob et al., 2018*; *Liberti et al., 2017*; *de Groot et al., 2020*; *Skocek et al., 2018*; *Scott et al., 2018*). The UCLA Miniscope project, a user-friendly miniature head-mounted microscope for in vivo calcium imaging in freely behaving animals, is one such project that has been accessible to a large number of users (*Aharoni et al., 2019*; *Cai et al., 2016*; *Shuman et al., 2020*; *Aharoni and Hoogland, 2019*).

With the increasing popularity of miniature microscopes, there is a growing need for analysis pipelines that can reliably extract neuronal activities from recording data. To address this need, numerous algorithms have been developed and made available to the neuroscience community. The principal component analysis or independent component analysis (PCA-ICA)-based approach (*Mukamel et al., 2009*) and region of interest (ROI)-based approach (*Cai et al., 2016*) were among the earliest algorithms that reliably detected the locations of neurons and extract their overall activities across pixels. However, one of the limitations of these approaches is that activities from cells that are spatially overlapping cannot be demixed. A subsequent constrained non-negative matrix factorization (CNMF) approach was shown to reliably extract neuronal activity from both two-photon and single-photon calcium imaging data (*Pnevmatikakis et al., 2016*), and demix the activities of overlapping cells. The CNMF algorithm models the video as a product of a 'spatial' matrix containing detected neuronal footprints (locations of cells) and a 'temporal' matrix containing the temporal calcium traces of each detected cell. This approach is particularly effective at addressing crosstalk between neurons, which is of particular concern in single-photon imaging, where the fluorescence from overlapping or nearby cells contaminates each other. Moreover, by deconvolving calcium traces, the CNMF algorithm enables a closer exploration of the underlying activity of interest, action potentials (*Friedrich et al., 2017*; *Vogelstein et al., 2010*). Originally developed for two-photon data, the CNMF algorithm did not include an explicit model of the out-of-focus fluorescence that is often present in single-photon miniature microscope recordings. This issue was addressed via the CNMF-E algorithm (*Zhou et al., 2018*), where a ring model is used as a background term to account for out-of-focus fluorescence. Later, an open-source Python pipeline for calcium imaging analysis, CaImAn, was published, which included both the CNMF and CNMF-E algorithms, as well as many other functionalities (*Giovannucci et al., 2019*). The latest development in analysis pipelines for in vivo miniature microscope data is MIN1PIPE (*Lu et al., 2018*), where a morphological operation is used to remove background fluorescence during preprocessing of the data, and a seed-based approach is used for initialization of the CNMF algorithm. Other approaches have also been used to extract signals from calcium imaging data, including an online approach (*Giovannucci et al., 2017*), $l0$ -penalization approach to infer spikes (*Jun et al., 2017*; *Jewell and Witten, 2018*), robust modeling of noise (*Inan et al., 2021*), and source detection using neural networks (*Klibisz et al., 2017*).

The open sharing of the algorithms necessary for the computation of neural activity has been exceptionally important for the field. However, implementation of these tools can be complex as many algorithms have numerous free parameters (those that must be set by the user) that can influence the outcomes, without clear guidance on how these parameters should be set or to what extent they affect results. Moreover, there is a lack of ground-truth data for in vivo miniature microscope imaging, making it hard to validate algorithms and/or parameters. Together, these obstacles make it challenging for neuroscience labs to adopt the analysis pipelines since it is difficult for researchers to adjust parameters to fit their data or to trust the output of the pipeline for downstream analysis. Thus,

the next challenge in open-source analysis pipelines for calcium imaging is to make the analysis tools more user-friendly and underlying algorithms more accessible to neuroscience researchers so that they can more easily understand the pipeline and interpret the results.

## Contributions of Minian

To increase the accessibility of the mathematical algorithms, transparency into how altering parameters alters the data output, and usability for researchers with limited computational resources and experience, we developed Minian (miniscope analysis), an open-source analysis pipeline for single-photon calcium imaging data inspired by previously published algorithms. We based Minian on the CNMF algorithm (*Giovannucci et al., 2019*; *Pnevmatikakis et al., 2016*), but also leverage methods from other pipelines, including those originally published by *Cai et al., 2016* and MIN1PIPE (*Lu et al., 2018*). To enhance compatibility with different types of hardware, especially laptops or personal desktop computers, we implemented an approach that supports parallel and out-of-core computation (i.e., computation on data that are too large to fit a computer's memory). We then developed interactive visualizations for every step in Minian and integrated these steps into annotated Jupyter Notebooks as an interface for the pipeline. We have included detailed notes and discussions on how to adjust the parameters from within the notebook and have included all free parameters in the code for additional flexibility. The interactive visualizations will help users to intuitively understand and visually inspect the effect of each parameter, which we hope will facilitate more usability, transparency, and reliability in calcium imaging analysis.

Minian contributes to three key aspects of calcium image data analysis:

1. *Visualization.* For each step in the pipeline, Minian provides visualizations of inputs and results. Thus, users can proceed step-by-step with an understanding of how the data are transformed and processed. In addition, all visualizations are interactive and support simultaneous visualization of the results obtained with different parameters. This feature provides users with knowledge about the corresponding outcome for each parameter value and allows the users to choose the outcome that fits best with their expectation. Hence, the visualizations also facilitate parameter exploration for each step, which is especially valuable when analyzing data from heterogeneous origins that may vary by brain region, cell type, species, and the extent of viral transfection.

2. *Memory demand.* One of the most significant barriers in adopting calcium imaging pipelines is the memory demand of algorithms. The recorded imaging data usually take up tens of gigabytes of space when converted to floating-point datatypes and often cannot fit into the RAM of standard computers without spatially and/or temporally downsampling. CaImAn (*Giovannucci et al., 2019*) addresses this issue by splitting the data into overlapping patches of pixels, processing each patch independently, and merging the results together. This enables out-of-core computation since at any given time only subsets of data are needed and loaded into memory. In Minian, we extend this concept further by flexibly splitting the data either spatially (split into patches of pixels) or temporally (split into chunks of frames). In this way, we avoid the need to merge the results based on overlapping parts. The result is a pipeline that supports out-of-core computation at each step, which gives nearly constant memory demand with respect to input data size. Minian can process more than 20 min of recording (approximately 12.6 GB of raw data) with 8 GB of memory, which makes Minian suitable to be deployed on modern personal laptops.

3. *Accessibility.* Minian is an open-source Python package. In addition to the codebase, Minian distributes several Jupyter Notebooks that integrate explanatory text with code and interactive visualizations of results. For each step in the notebook, detailed instructions, as well as intuition about the underlying mathematical formulation, are provided, along with code, which can be directly executed from within the notebook. Upon running a piece of code within the notebook, visualizations appear directly below. In this way, the notebooks serve as a complement to traditional API documentations of each function. In addition, users can easily rearrange and modify the pipeline notebook to suit their needs without diving into the codebase and modifying the underlying functions. The notebooks distributed by Minian can simultaneously function as a user guide, template, and production tool. We believe the inclusion of these notebooks, in combination with Minian's other unique features, can increase understanding of the underlying functioning of the algorithms and greatly improve the accessibility of miniature microscopy analysis pipelines.

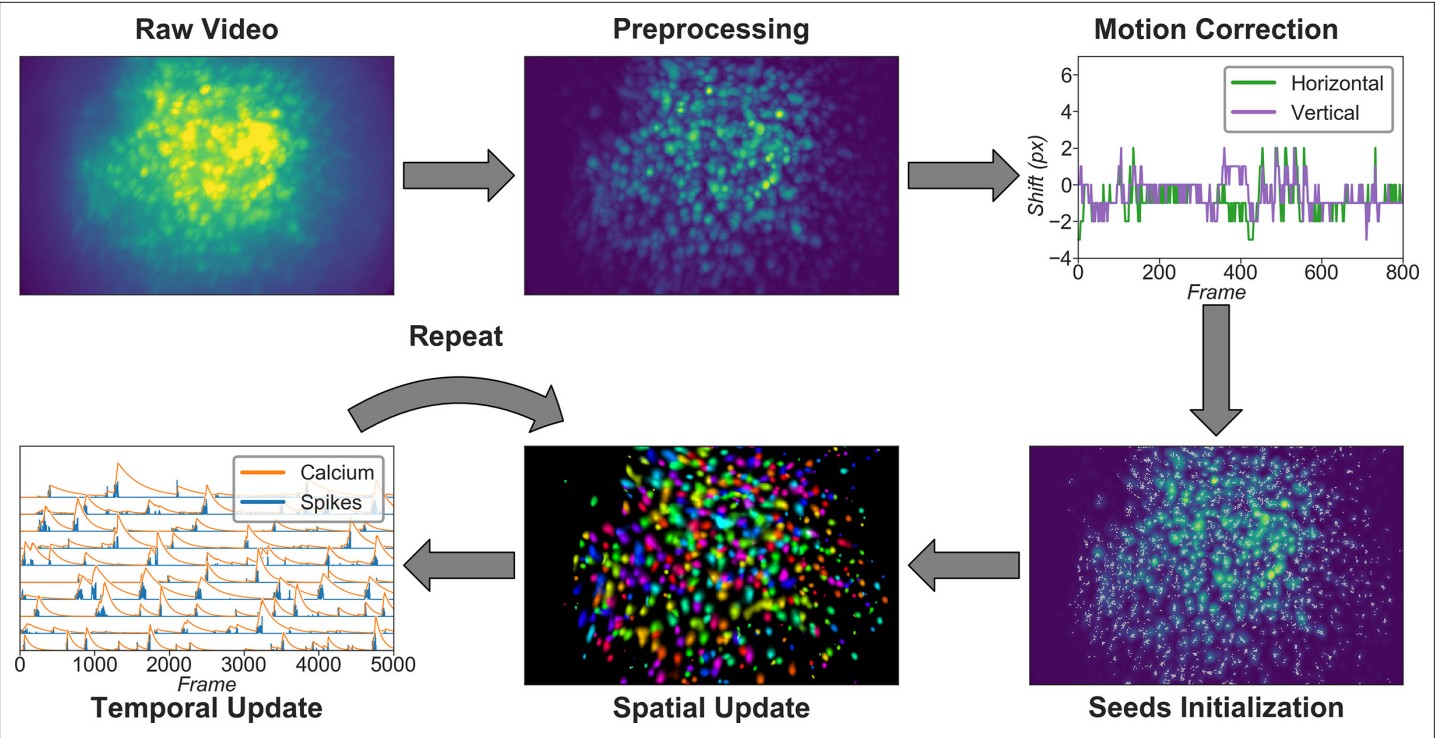

**Figure 1.** Overview of the analysis pipeline. The analysis is divided into five stages: preprocessing, where sensor noise and background fluorescence from scattered light are removed; motion correction, where rigid motion of the brain is corrected; seeds initialization, where the initial spatial and temporal matrices for later steps are generated from a seed-based approach; spatial update, where the spatial footprints of cells are further refined; and temporal update, where the temporal signals of cells are further refined. The last two steps of the pipeline are iterative and can be repeated multiple times until a satisfactory result is reached.

### Article organization

This article is organized as follows. Since Minian's major contribution is usability and accessibility, we first present the detailed steps in the analysis pipeline in 'Materials and methods'. Following a step-by-step description of the algorithms Minian adapted from existing works, we present novel visualizations of the results, as well as how users can utilize these visualizations. In 'Results', we benchmark Minian across two brain regions and show that spatial footprints and the temporal activity of cells can be reliably extracted. We also show that the cells extracted by Minian in hippocampal CA1 exhibit stable spatial firing properties consistent with the existing literature.

## Materials and methods

Here, we present a detailed description of Minian. We begin with an overview of the Minian pipeline. Then, we provide an explanation of each step, along with the visualizations. Lastly, we provide information regarding hardware and dependencies.

### Overview of Minian

Minian comprises five major stages, as shown in **Figure 1**. Raw videos are first passed into a preprocessing stage. During preprocessing, the background caused by vignetting (in which the central portion of the field of view is brighter) is corrected by subtracting a minimum projection of the movie across time. Sensor noise, evident as granular specks, is then corrected with a median filter. Finally, background fluorescence is corrected by the morphological process introduced in MIN1PIPE (*Lu et al., 2018*). The preprocessed video is then motion-corrected with a standard template-matching algorithm based on cross-correlation between each frame and a reference frame (*Brunelli, 2009*). The motion-corrected and preprocessed video then serves as the input to initialization and CNMF algorithms. The seed-based initialization procedure looks for local maxima in max projections of different

subsets of frames and then generates an over-complete set of seeds, which are candidate pixels for detected neurons. Because this process is likely to produce many false positives, seeds are then further refined based on various metrics, including the amplitude of temporal fluctuations and the signal-to-noise ratio of temporal signals. The seeds are transformed into an initial estimation of cells' spatial footprints based on the correlation of neighboring pixels with each seed pixel, and the initial temporal traces are in turn estimated based on the weighted temporal signal of spatial footprints. Finally, the processed video, initial spatial matrix, and temporal matrix are fed into the CNMF algorithm. The CNMF algorithm first refines the spatial footprints of the cells (spatial update). The algorithm then denoises the temporal traces of each cell while simultaneously deconvolving the calcium trace into estimated 'spikes' (temporal update). CNMF spatial and temporal updates are performed iteratively and can be repeated until a satisfactory result is reached through visual inspection. Typically, this takes two cycles of spatial, followed by temporal, updates. Minian also includes a demo dataset that allows the user to run and test the pipeline comprised of the pre-made Jupyter Notebook immediately after installation.

## Setting up

The first section in the pipeline includes house-keeping scripts to import packages and functions, defining parameters, and setting up parallel computation and visualization. Most notably, the distributed cluster that carries out all computations in Minian are set up in this section. By default, the cluster runs locally with multicore CPUs; however, it can be easily scaled up to run on distributed computers. The computation in Minian is optimized such that in most cases the memory demand for each process/core can be as low as 2 GB. However, in some cases depending on the hardware, the state of operating system and data locality, Minian might need more than 2 GB per process to run. If a memory error (KilledWorker) is encountered, it is common for users to increase the memory limit of the distributed cluster to get around the error. Regardless of the exact memory limit per process, the total memory usage of Minian roughly scales linearly with the number of parallel processes. The number of parallel processes and memory usage of Minian is completely limited and managed by the cluster configuration, allowing users to easily change them to suit their needs.

## Preprocessing

### Loading data and downsampling

Currently Minian supports .avi movies, the default output from the UCLA Miniscopes, and .tif stacks, the default output from Inscopix miniscopes. This functionality can be easily extended to support more formats if desired. Users are required to organize their data so that each recording session is contained in a single folder. Because Minian can extract relevant metadata from folder nomenclature (e.g., animal name, group, date), we suggest organizing the video folders based upon animal and other experiment-related groupings to facilitate the incorporation of metadata into Minian output files.

Minian supports downsampling on any of the three video dimensions (height, width, and frames). Two downsampling strategies are currently implemented: either subsetting data on a regular interval or calculating a mean for each interval. At this stage, users are required to specify (1) the path to their data, (2) a pattern of file names to match all the videos to be processed (e.g., all files containing 'msCam,' a typical pattern resulting from Miniscope recordings), (3) a Python dictionary specifying whether and how metadata should be pulled from folder names, (4) another Python dictionary specifying whether and on which dimension downsampling should be carried out, and (5) the downsampling strategy, if desired.

Once specified, the data can be immediately visualized through an interactive viewer, as shown in *Figure 2*. Along with a player to visualize every frame in the video, the viewer also plots summary values such as mean, maximum, or minimum fluorescence values across time. This helps users to check their input data and potentially exclude any artifacts caused by technical faults during experiments (e.g., dropped frames). Users can further subset data to exclude specified frames, if necessary. Finally, restricting the analysis to a certain subregion of the field of view during specific steps could be beneficial. For example, if the video contains anchoring artifacts resulting from dirt on the lenses, it is often better to avoid such regions during motion correction. To facilitate this, the viewer provides a feature

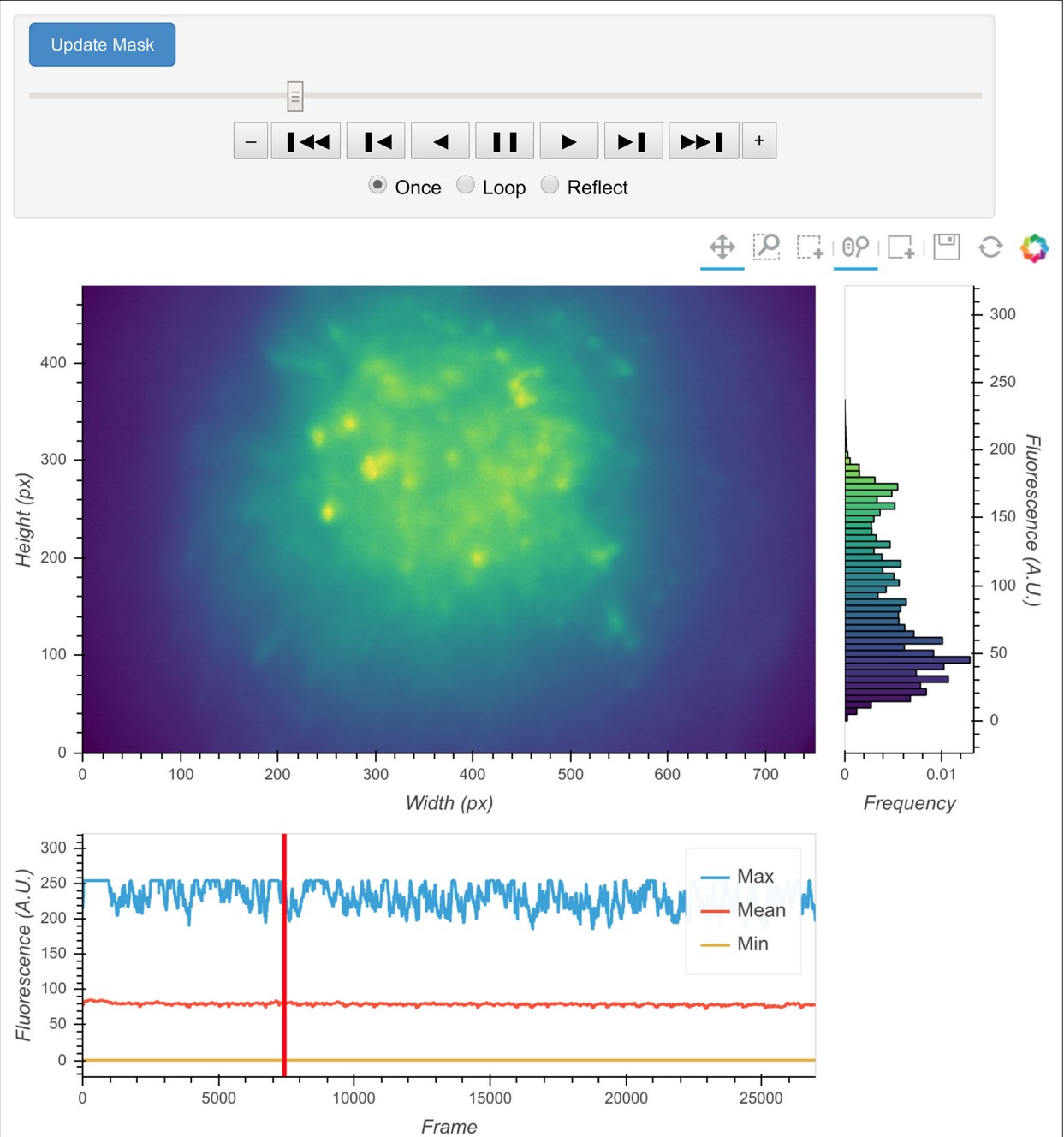

**Figure 2.** Interactive visualization of raw input video. One frame is shown in the central panel of the visualization that can be interactively updated with the player toolbar on the top. A histogram of fluorescence intensity of the current frame is shown on the right and will update in response to zooming in on the central frame. A line plot of summary values across time is shown at the bottom. Here, the maximum, mean, and minimum fluorescence values are plotted. These summaries are useful in checking whether there are unexpected artifacts or gaps in the recording. Finally, the user can draw an arbitrary box in the central frame, and the position of this boxed region can be recorded and used as a mask during later steps. For example, during motion correction a subregion of the data containing a stable landmark might provide better information on the motion.

where users can draw an arbitrary box within the field of view and have it recorded as a mask. This mask can be passed into later motion correction steps to avoid the biases resulting from the artifacts.

## Vignetting correction

Single-photon miniature microscope data often suffer from a vignetting effect in which the central portion of the field of view appears brighter than the periphery. Vignetting is deleterious to subsequent

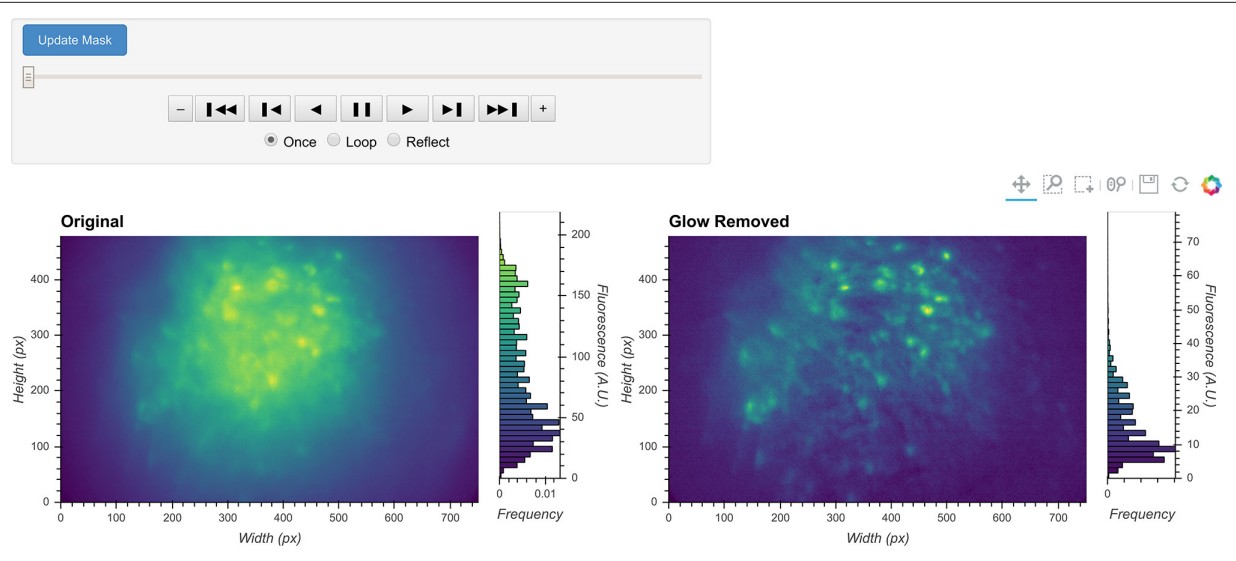

**Figure 3.** General visualization of preprocessing. The same visualization of input video can be used to visualize the whole video before and after specific preprocessing steps side-by-side. The effect of vignetting correction is visualized here. The image and accompanying histogram on the left side show the original data; the data after vignetting correction are shown on the right side. Any frame of the data can be selected with the player toolbar and histograms are responsive to all updates in the image.

processing steps and should be removed. We find that the effect can be easily extracted by taking the minimum fluorescence value across time for each pixel and subtracting this value from each frame, pixel-wise. One of the additional benefits of subtracting the minimum is that it preserves the raw video's linear scale.

The result of this step can be visualized with the same video viewer used in the previous step. In addition to visualizing a single video, the viewer can also show multiple videos side-by-side (e.g., the original video and the processed video), as shown in *Figure 3*. The operation/visualization is carried out 'on-the-fly' upon request for each frame, and users do not have to wait for the operation to finish on the whole video to view the results.

## Denoising

Next, we correct for salt-and-pepper noise on each frame, which usually results from electronic pixel noise. By default, we pass each frame through a median filter, which is generally considered particularly effective at eliminating this type of noise, though other smoothing filters like Gaussian filters and anisotropic filters can also be implemented. The critical parameter here is the window size of the median filter. A window size that is too small will make the filter ineffective at correcting outliers, while a window size that is too large will remove finer gradient and edges that are much smaller than the window size, and can result in a failure to distinguish between adjacent cells.

The effect of the window size can be checked with an interactive visualization tool used across the preprocessing stage, as shown in *Figure 4*. Additionally, here we show an example of the effect of window size on the resulting data in *Figure 5*. Users should see significantly reduced amount of salt-and-pepper noise in the images, which should be made more obvious by the contour plots. At the same time, users should keep the window size below the extent where over-smoothing occurs. As a heuristic, the average cell radius in pixel units works well since a window of the same size as an average cell is unlikely to blend different cells together, while still being able to adequately smooth the image.

## Morphological background removal

Next, we remove any remaining background presumably caused by out-of-focus and tissue fluorescence. To accomplish this, we estimate the background using a morphological opening process first introduced for calcium imaging analysis in MIN1PIPE (*Lu et al., 2018*), which acts as a size filter that removes cell bodies. The morphological opening is composed of two stages: erosion followed

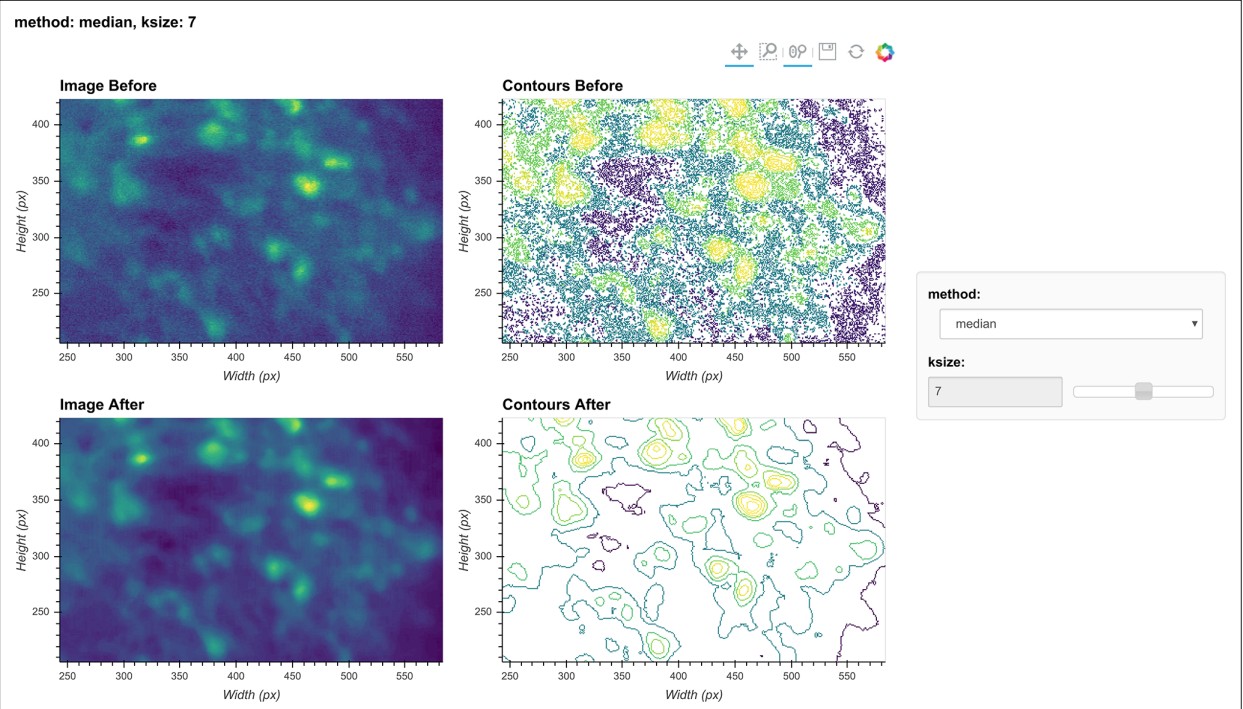

**Figure 4.** Visualization of denoising. Here, a single frame from the data is passed through the background removal, and both the image and a contour plot are shown for the frame before and after the process. The contour plots show the iso-contour of five intensity levels spaced linearly across the full intensity range of the corresponding image. The plots are interactive and responsive to the slider of the window size on the right, thus the effect of different window sizes for denoising can be visualized.

by dilation. In morphological erosion, the image is passed through a filter where each pixel will be substituted by the minimum value within the filter window. The effect of this process is that any bright 'feature' that is smaller than the filter window will be 'eroded' away. Then, the dilation process accomplishes the reverse by substituting each pixel with the maximum value in the window, which 'dilates' small bright features to the extent of the filter window size. The combined effect of these two stages is that any bright 'feature' that is smaller than the filter window is removed from the image. If we choose

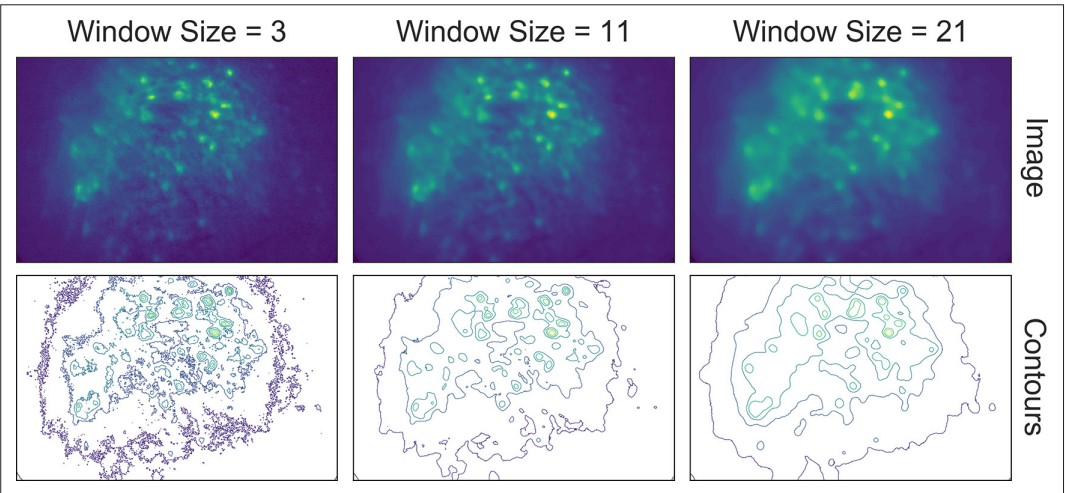

**Figure 5.** Effect of window size on denoising. One example frame is chosen from the data, and the resulting images (top row) and contour plots (bottom row) are shown to demonstrate the effect of window size on denoising. Here, a window size of 11 (middle column) is appropriate while both smaller and larger window sizes result in artifacts.

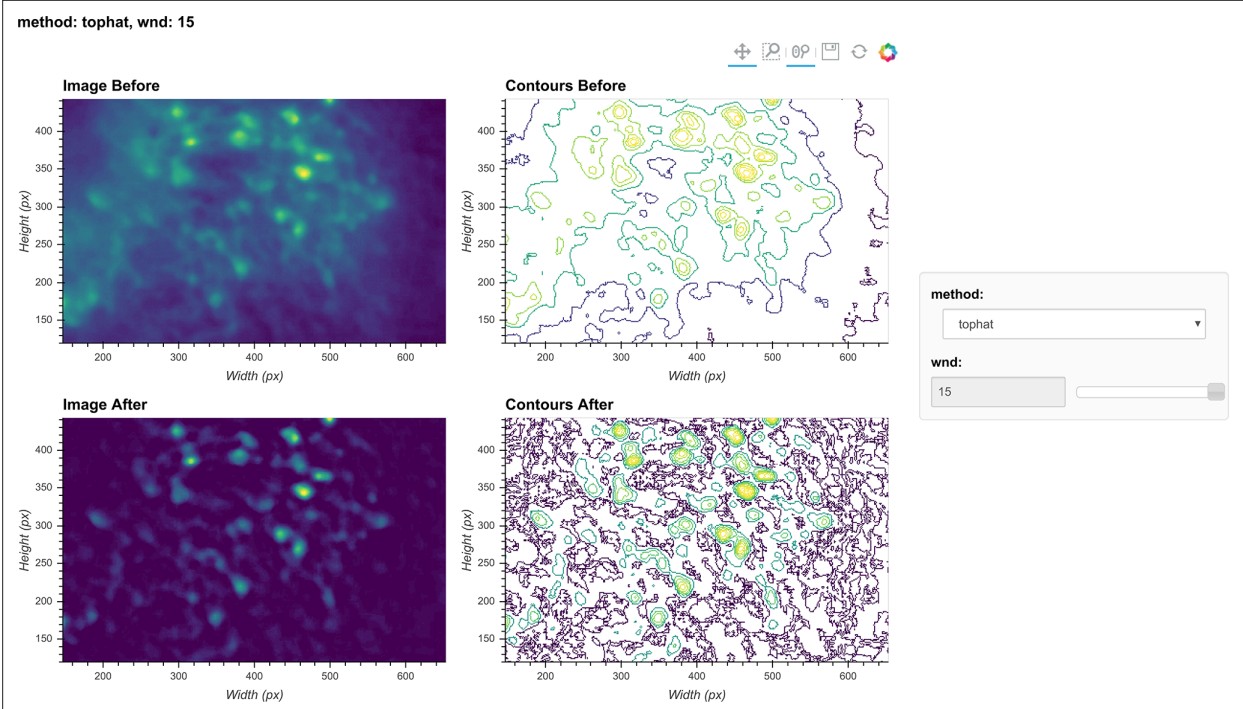

**Figure 6.** Visualization of background removal. Here, a single frame from the data is passed through background removal, and both the image and a contour plot are shown for the frame before and after the process. The plots are interactive and responsive to the slider of the window size on the right, thus the effect of different window sizes for background removal can be visualized.

the window size to match the expected cell diameter, performing a morphological opening will likely remove cells and provide a good estimation of background. Hence, each frame is passed through the morphological opening operation and the resulting image is subtracted from the original frame.

Although the window size parameter for the morphological opening can be predetermined by the expected cell diameter, it is helpful to visually inspect the effect of morphological background

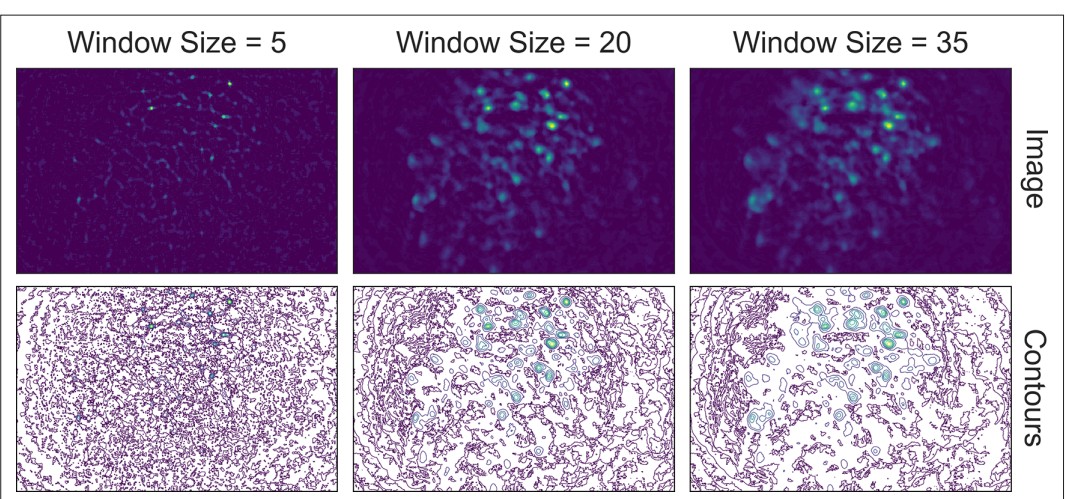

**Figure 7.** Effect of window size on background removal. One example frame is chosen from the data, and the resulting images (top row) and contour plots (bottom row) are shown to demonstrate the effect of window size on background removal. The contour plots show the iso-contour of five intensity levels spaced linearly across the full intensity range of the corresponding image. Here, a window size of 20 pixels (middle column) is appropriate while both smaller and larger window sizes produce unsatisfactory results: a window size too small (left column) artificially limits the size of cells, and a window size too large (right column) does not remove the background effectively.

removal. The effect of different window sizes can be visualized with the same tool used in denoising, as shown in *Figure 6*. Additionally, here we show an example of the effect of window size on the resulting data in *Figure 7*. In this case, a window size of 20 pixels is considered appropriate because the resulting cells are appropriately sized and sharply defined. In contrast, a smaller window results in limiting both the size and intensity of the cells. On the other hand, residual out-of-focus fluorescence becomes visible when the window size is set too large.

## Motion correction
### Estimate and apply translational shifts

We use a standard template-matching algorithm based on cross-correlation to estimate and correct for translational shifts (*Brunelli, 2009*). In practice, we found that this approach is sufficient to correct for motion artifacts that could have a significant impact on the final outcome. Briefly, for a range of possible shifts, a cross-correlation between each frame and a template frame is calculated. The shift producing the largest cross-correlation is estimated to reflect the degree of movement from the template and is corrected by applying a shift to the frame in that direction. We apply this operation to the whole movie in a divide-and-conquer manner. We split the movie into chunks of frames, within which we register both the first and last frames to the middle frame. We then take the max projections of the three frames that have been registered in each chunk and group every three chunks together and register them using the max projections as templates. After the registration, the three chunks that have been registered are treated as a new single chunk and we again take the max projection to use as a template for further registration. In this way, the number of frames registered in each chunk keeps increasing in powers of 3 (3, 9, 27, 81, etc.), and we repeat this process recursively until all the frames are covered in a single chunk and the whole movie is registered. Since the motion correction is usually carried out after background removal, we essentially use cellular activity as landmarks for registration. Sometimes this can be problematic when cellular activity is very sparse and different across two chunks (e.g., when only two different cells fired in two chunks), leading to false estimation of shifts. To overcome this problem, every time shift is estimated using a max projection from two chunks, we also estimate a shift with the two consecutive frames bordering the chunks (i.e., the last frame from the earlier chunk and the first frame from the latter chunk). In most cases, the shifts estimated with these two sets of templates should be close, in which case we use the shifts estimated with the max projection as the final output. However, when the two estimated shifts differ too much from each other, we use the shifts estimated with consecutive frames as the final output. The reason we still favor using max projections in most cases is that registering with consecutive frames can lead to very fast accumulation of error and a slow drifting artifact in the estimated shifts. In practice, we find that such a process can account for almost all motion in the brain, so currently we only implemented estimation of translational shifts. If the user would like to take advantage of anatomical landmarks (such as blood vessels) within the field of view and would like to implement motion correction before all background subtraction steps have been performed, the pipeline can be easily modified to do so. After the estimation of shifts, the shift in each direction is plotted across time and visualization of the data before and after motion correction is displayed in Minian (see *Figure 1*, top right).

## Seed initialization
### Generation of an over-complete set of seeds

The CNMF algorithm is a powerful approach to extract cells' spatial structure and corresponding temporal activity. However, the algorithm requires an initial estimate of cell locations/activity, which it then refines. We use a seed-based approach introduced in MIN1PIPE (*Lu et al., 2018*) to initialize spatial and temporal matrices for CNMF. The first step is to generate an over-complete set of seeds, representing the potential centroids of cells. We iteratively select a subset of frames, compute a maximum projection for these frames, and find the local maxima on the projections. This workflow is repeated multiple times, and we take the union of all local maxima across repetitions to obtain an over-complete set of seeds. In this way, we avoid missing cells that only fire in short periods of time that might be masked by taking a maximum projection across the whole video.

During seed initialization, the first critical parameter is the spatial window for defining local maxima. Intuitively, this should be the expected diameter of cells. The other critical parameter is an intensity threshold for a local maximum to be considered a seed. Since the spatial window for local maxima is

small relative to the field of view, a significant number of local maxima are usually false positives and do not actually reflect the location of cells. Thresholding the fluorescence intensity provides a simple way to filter out false local maxima, and usually a very low value is enough to produce satisfactory results. We have found a value of 3 usually works well (recall that the range of fluorescence intensity is usually 0–255 for unsigned 8-bit data). An alternative strategy to thresholding the intensity is to model the distribution of fluorescence fluctuations and keep the seeds with relatively higher fluctuations. This process is described in 'Seeds refinement with a Gaussian mixture model' and is accessible if the user prefers explicit modeling over thresholding.

Finally, the temporal sampling of frames for the maximum projections also impacts the result. We provide two implementations here: either taking a rolling window of frames across time or randomly sampling frames for a user-defined number of iterations. For the rolling window approach, users can specify a temporal window size (the number of successive frames for each subset) and a step size (the interval between the start of subsets). For the random approach, users can specify the number of frames in each subset and the total number of repetitions. We use the rolling window approach as the default.

The resulting seeds are visualized on top of a maximum projection image (plot not shown). Although the spatial window size of local maxima can be predetermined, the parameters for either the rolling window or random sampling of frames are hard to estimate intuitively. We provide default parameters that generally provide robust results. However, the user is also free to vary these parameters to obtain reasonable seeds. As long as the resulting seeds are not too dense (populating almost every pixel) or too sparse (missing cells that are visible in the max projection), subsequent steps can be performed efficiently and are fairly tolerable to the specific ways the seeds are initialized.

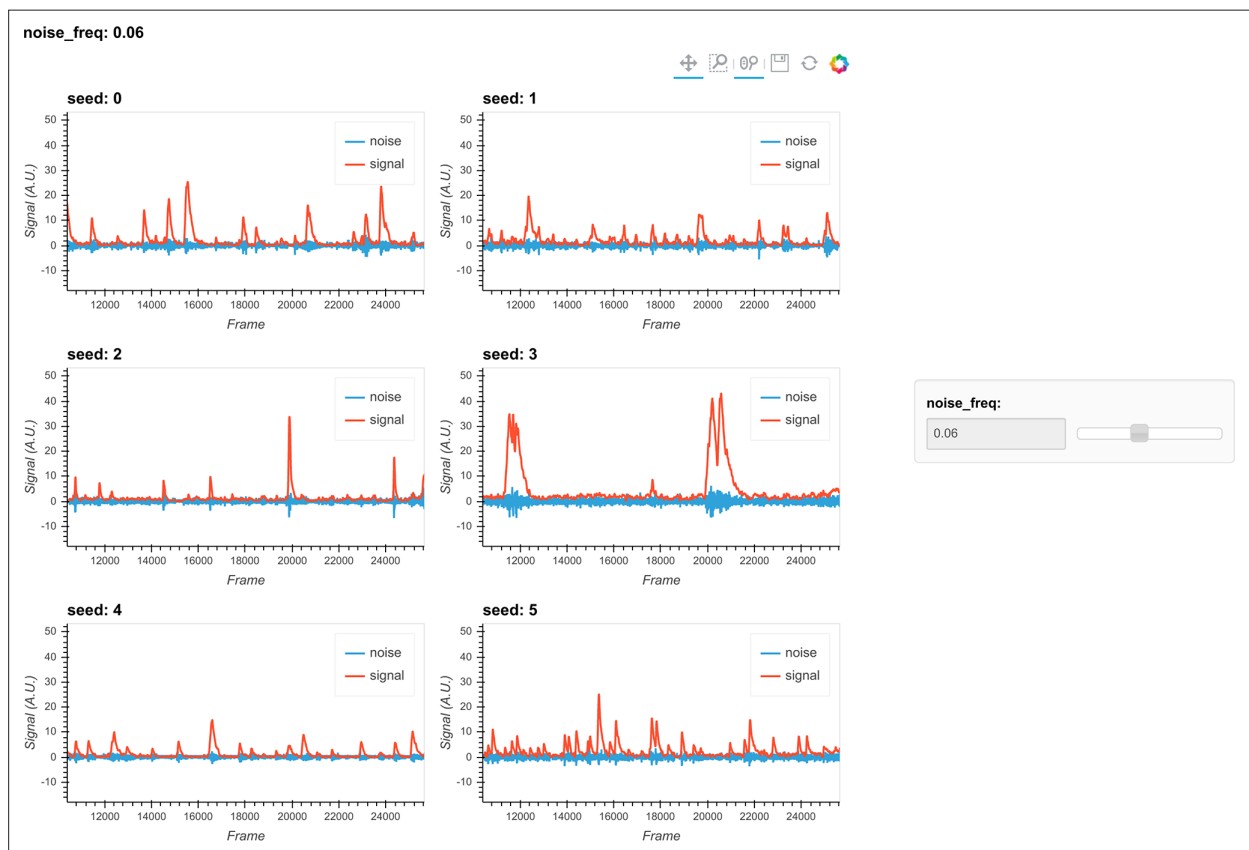

**Figure 8.** Visualization of noise frequency cutoff. The cutoff frequency for noise is one of the critical parameters in the pipeline that affects both the seed initialization process and constrained non-negative matrix factorization's (CNMF's) temporal update steps. Here, we help the user determine that parameter by plotting temporal traces from six example seeds. In each plot, the raw signal is passed through a high-pass and low-pass filter at the chosen frequency, and the resulting signals are plotted separately as 'noise' and 'signal.' The plots are responsive to the chosen frequency controlled by the slider on the right. In this way, the user can visually inspect whether the chosen frequency can effectively filter out high-frequency noise without deforming the calcium signal.

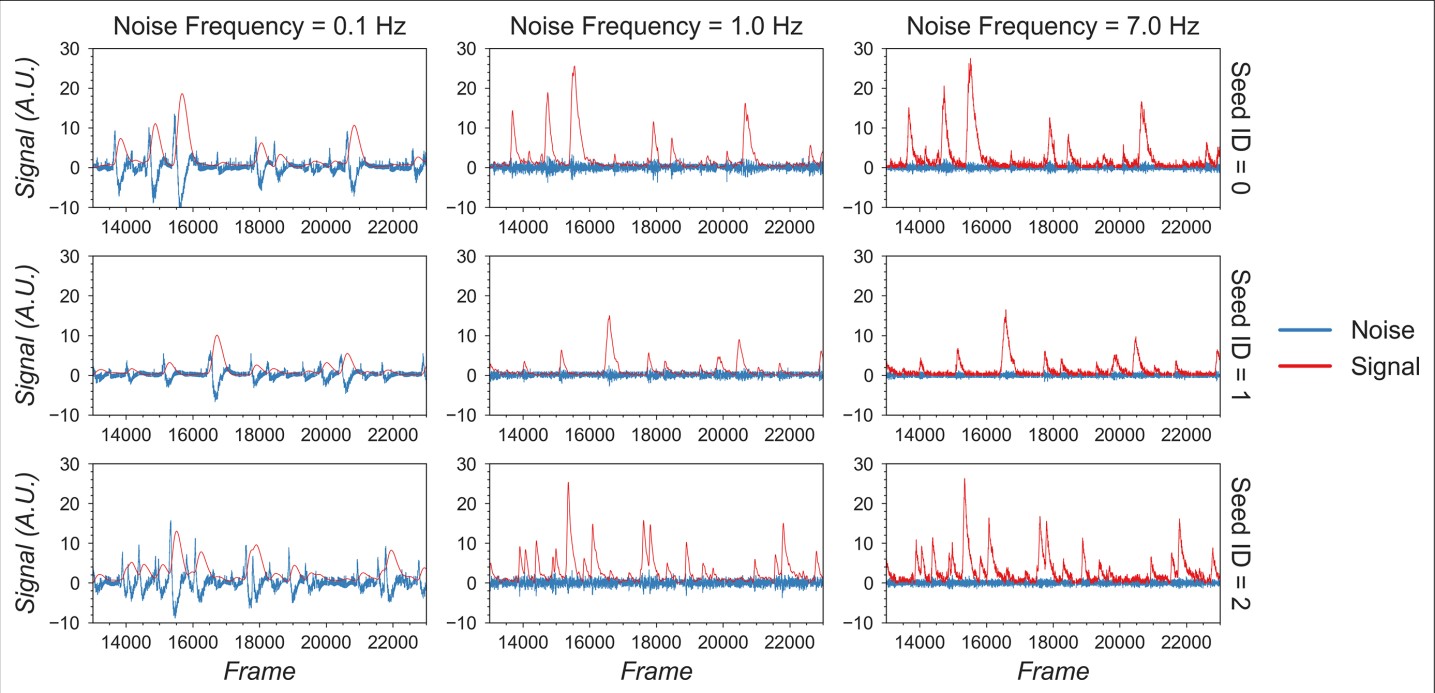

**Figure 9.** Example of filtered traces with different frequency cutoffs. Here, the temporal dynamics of three example seeds are chosen, and the low-pass and high-pass filtered traces with different frequency cutoffs are shown. The low-pass filtered trace corresponds to 'signal,' while the high-pass filtered trace corresponds to 'noise.' Here, a 1 Hz cutoff frequency is considered appropriate since calcium dynamics and random noise are cleanly separated. A cutoff frequency smaller than 1 Hz left the calcium dynamics in the 'noise' trace, while a cutoff frequency larger than 1 Hz let random noise bleed into the 'signal' trace (i.e., high-frequency fluctuations are presented in periods where the cells seem to be inactive).

## Refinement with peak-to-noise ratio

Next, we refine the seeds by looking at what we call the peak-to-noise ratio of the temporal traces and discard seeds with low peak-to-noise ratios. To compute this ratio, we first separate the noise from the presumed real signal. Calcium dynamics are mainly composed of low-frequency fluctuations (from the slow kinetics of the calcium fluctuations) while noise is composed of higher frequency fluctuations. Thus, to separate the noise from the calcium dynamics we pass the fluorescence time trace of each seed through a low-pass and a high-pass filter to obtain the 'signal' and 'noise' of each seed. We then compute the difference between the maximum and minimum values (or peak-to-peak values) for both 'signal' and 'noise,' and the ratio between the two difference values defines the peak-to-noise ratio. Finally, we filter out seeds whose peak-to-noise value falls below a user-defined threshold.

The first critical parameter here is the cutoff frequency that separates 'signal' from 'noise.' This parameter is also important for subsequent steps when implementing the CNMF algorithm. We provide a visualization tool, shown in *Figure 8*, to help users determine cutoff frequency. In the visualization, six seeds are randomly selected, and their corresponding 'signal' and 'noise' traces are plotted. The user is then able to use a dynamic slider on the right side of the plots to adjust the cutoff frequency and view the results. The goal is to select a frequency that best separates signal from noise. A cutoff frequency that is too low will leave true calcium dynamics absorbed in 'noise' (*Figure 9*, left panel), while a frequency that is too high will let 'noise' bleed into 'signal' (*Figure 9*, right panel). A suitable frequency is therefore the one where the 'signal' captures all of the characteristics of the calcium indicator dynamics (i.e., large, fast rise, and slow decay), while the 'noise' trace remains relatively uniform across time (*Figure 9*, middle panel). The interactive plots make this easy to visualize. We also provide an example in *Figure 9* to show how cutoff frequency influences the separation of 'signal' from 'noise.' The second parameter is the threshold of peak-to-noise ratio value. In practice, we have found a threshold of 1 works well in most cases. An additional advantage of using 1 is that it reflects the intuitive interpretation that fluctuations in a real 'signal' should be larger than fluctuations in 'noise.'

## Refinement with Kolmogorov–Smirnov tests

Finally, we refine the seeds with a Kolmogorov–Smirnov test. The Kolmogorov–Smirnov test assesses the equality of two distributions and can be used to check whether the fluctuation of values for each seed is non-normally distributed. We expect the noisy fluorescence values when a cell is not firing to form a Gaussian distribution with small mean value, and the fluorescence values when a cell is firing should have a much higher mean value and frequency than expected by the null Gaussian distribution. Therefore, seeds corresponding to cells should be non-normally distributed. We use a default significance threshold of 0.05. In some cases, this might be too conservative or too liberal. Users can tweak this threshold or skip this step altogether depending on the resulting seeds.

## Merge seeds

There will usually be multiple seeds for a single cell, and it is best to merge them whenever possible. We implement two criteria for merging seeds: first, the distance between the seeds must be below a given threshold, and second, the correlation coefficient of the temporal traces between seeds must be higher than a given threshold. To avoid bias in the correlation due to noise, we implement a smoothing operation on the traces before calculating the correlation. The critical parameters are the distance threshold, correlation threshold, and cutoff frequency for the smoothing operation. While the distance threshold is arbitrary and should be explored, often the average radius of cells provides a good starting point. The cutoff frequency should be the same as that used during the peak-to-noise ratio refinement described above, and the correlation should be relatively high (we typically use 0.8, but this can be refined by the user). The resulting merged seeds can be visualized on the max projection. Since the main purpose of this step is to alleviate computation demands for downstream steps, it is fine to have multiple seeds for a single visually distinct cell. However, users should make sure each of the visually distinct cells still has at least one corresponding seed after the merge.

## Initialize spatial and temporal matrices from seeds

The last step before implementing CNMF is to initialize the spatial and temporal matrices for the CNMF algorithm from the seeds. These matrices are generated with one dimension representing each putative cell and the other representing each pixel or time, respectively. In other words, the spatial matrix represents the spatial footprint for each cell at each pixel location and the temporal matrix represents the temporal fluorescence value of each cell on each frame. We assume each seed is the center of a potential cell, and we first calculate the spatial footprint for each cell by taking the cosine similarity between the temporal trace of a seed and the pixels surrounding that seed. In other words, we generate the weights in the spatial footprint by computing how similar the temporal activities of each seed are to the surrounding pixels. Then, we generate the temporal activities for each potential cell by taking the input video and weighting the contribution of each pixel to the cell's temporal trace by the spatial footprint of the cell. The final products are a spatial matrix and temporal matrix.

Besides the two matrices representing neuronal signals, there are two additional terms in the CNMF model that account for background fluorescence modeled as a spatial footprint for the background and a temporal trace of background activity. To estimate these terms, we subtract the matrix product of our spatial and temporal matrices, which represent cellular activities, from the input data. We take the mean projection of this remainder across time as an estimation of the spatial footprint of the background, and we take the mean fluorescence for each frame as the temporal trace of the background.

Users can tweak two parameters to improve the outcome and performance of this step: a threshold for cosine similarity and a spatial window identifying pixels on which to perform this computation. To keep the resulting spatial matrix sparse and keep irrelevant pixels from influencing the temporal traces of cells, we set a threshold for the cosine similarity of temporal traces compared to the seed, where pixels whose similarity value falls below this threshold will be set to zero in the spatial footprint of the cell. Cosine similarity is, in essence, a correlation (the scale is 0–1) and thresholds of 0.5 and higher work well in practice. Computing many pairwise similarity measurements is computationally expensive, and it is unnecessary to compute the similarities between pixels that are far apart because they are unlikely to have originated from the same cell. We therefore set a window size to limit the number of pixel pairs to be considered. This size should be set large enough so that it does not limit the size of spatial footprints, but not unnecessarily large to the extent where it will impact performance. In practice, a window size equal to the maximum expected cell diameter is reasonable.

## Constrained non-negative matrix factorization

### Estimate spatial noise

CNMF requires that we first estimate the spatial noise over time for each pixel in the input video. The spatial noise of each pixel is simply the power of the high-frequency signals in each pixel. The critical parameter here is again the cutoff frequency for 'noise,' and users should employ the visualization tools as described above during peak-to-noise ratio refinement to determine this frequency (see 'Refinement with peak-to-noise ratio).

### Spatial update

Next, we proceed to the spatial update of the CNMF algorithm. The original paper describing this algorithm (*Pnevmatikakis et al., 2016*) contains a detailed theoretical derivation of the model. Here, we provide only a conceptual overview of the process so that users can understand the effect of each parameter. The CNMF framework models the input video to be the product of the spatial and temporal matrices representing signals contributed by real cells, a background term, and random noise. In equation form, this is $\mathbf{Y} = \mathbf{AC} + \mathbf{B} + \mathbf{E}$, where $\mathbf{Y}$ represents the input video, $\mathbf{A}$ represents the spatial matrix containing the spatial footprints for all putative cells, $\mathbf{C}$ represents the temporal matrix containing the calcium dynamics for all putative cells, $\mathbf{B}$ represents the spatial-temporal fluctuation of background, and $\mathbf{E}$ represents error or noise. Since the full problem of finding proper $\mathbf{A}$ and $\mathbf{C}$ matrices is hard (nonconvex), we break down the full process into spatial update and temporal update steps, where iterative updates of $\mathbf{A}$ and $\mathbf{C}$ are carried out, respectively. Each iteration will improve on previous results and eventually converge on the best estimation.

During the spatial update, given an estimation of the temporal matrix and the background term, we seek to update the spatial matrix so that it best fits the input data, along with the corresponding temporal traces. To do so, we first subtract the background term from the input data so that the remainder is composed only of signals from cells and noise. Then, for each pixel, the algorithm attempts to find the weights for each cell's spatial footprint that best reproduces the input data ($\mathbf{Y}$) with the constraint that individual pixels should not weigh on too many cells (controlled through what is called a sparseness penalty). To reduce computational demand, we do this for each pixel independently and in parallel to improve performance, while retaining the 'demixing' power of the CNMF algorithm by updating the weights for all cells simultaneously. In the optimization process, the function to be minimized contains both a squared error term to assess error, and an $\ell 1$-norm term to promote sparsity (*Giovannucci et al., 2019*). The optimization process can be expressed formally as

$$\underset{\mathbf{A},\mathbf{b}}{\text{minimize}} \quad \|\mathbf{Y}(p,:) - \mathbf{A}(p,:)\mathbf{C} - \mathbf{bf}\| + \lambda\|\mathbf{A}(p,:)\|_1$$

$$\text{subject to} \quad \mathbf{A}, \mathbf{b} \geq 0$$

where $\mathbf{Y}(p,:)$ denotes the input movie data indexed at $p$th pixel, $\mathbf{A}(p,:)$ denotes the spatial matrix indexed at $p$th pixel across all putative cells, and $\mathbf{C}$, $\mathbf{b}$, $\mathbf{f}$ denote the temporal matrix, spatial footprint of background term, and temporal fluctuation of background term, respectively. The scalar $\lambda$ represents the sparse penalty that controls the balance between the error term and sparsity term.

Lastly, the spatial footprint of the background term is updated in the exact same way, together with other putative cells. However, the background term the temporal activity used in the spatial update is not constrained by the autoregressive model. After the spatial footprint of the background term is updated, we subtract the neural activity ($\mathbf{AC}$) from the input data to get residual background fluctuations. Then, the temporal activity of background term is calculated as the projection of residual onto the new background spatial footprint, where the raw activities of each pixel are weighted by the spatial footprint.

In other CNMF implementations, the estimated spatial noise is used to determine the scaling of the $\ell 1$-norm term in the target function and control the balance between error and sparsity of the result. However, in practice we find that it does not always give the best result for all types of datasets. For example, sometimes the estimated spatial noise is too large, which results in an overly conservative estimation of spatial footprints. Hence, we have introduced a sparseness penalty on top of the estimated scaling factor for the $\ell 1$-norm term. This parameter gives users more control over how sparsity should be weighted in the updating process. The higher the number, the higher the penalty imposed by the $\ell 1$-norm, and the more sparse the spatial footprints will become. The effect of this parameter

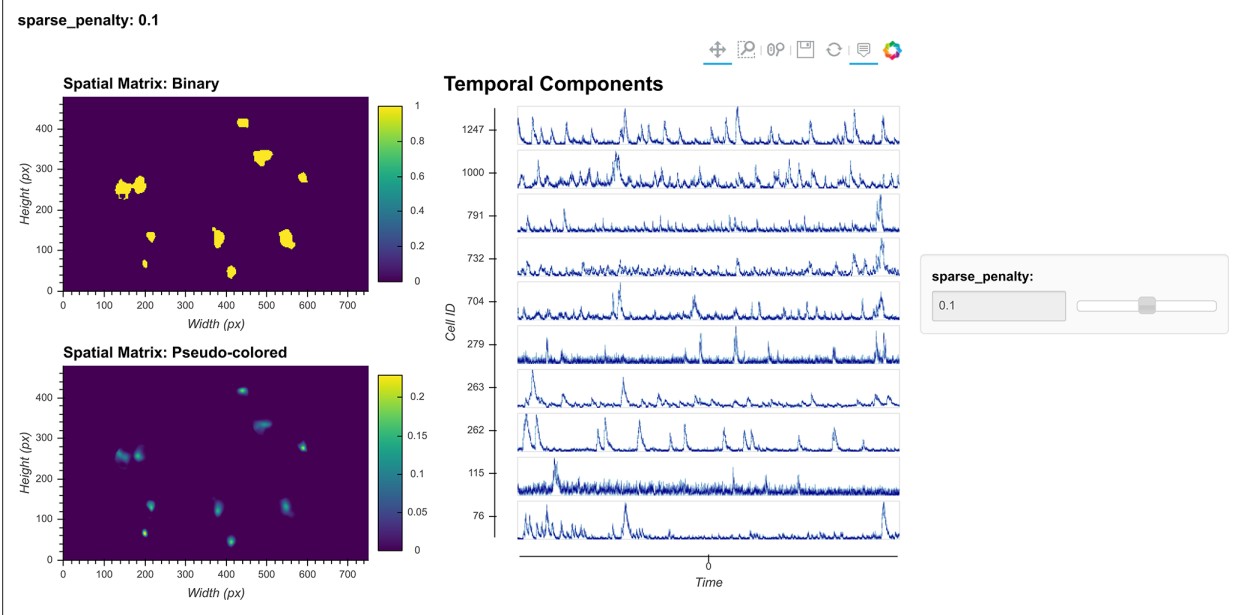

**Figure 10.** Visualization of spatial updates. Here, 10 cells are randomly chosen to pass through spatial update with different parameters. The resulting spatial footprints, as well as binarized footprints, are plotted. In addition, the corresponding temporal traces of cells are plotted. The user can visually inspect the size and shape of the spatial footprints and at the same time easily determine whether the results are sparse enough by looking at the binarized footprints.

can be visualized with the tool shown in *Figure 10*. Users can employ this tool to determine the best sparseness penalty for their data, where the binarized spatial footprint representing nonzero terms should approach the visible part of the spatial footprint as much as possible, without reducing the amplitude of spatial footprints to the extent that cells are discarded in the spatial update. *Figure 11* shows an example of the effect of changing the sparseness penalty on the resulting spatial footprints. A sparseness penalty of 0.1 is considered appropriate in this case. When the sparseness penalty is

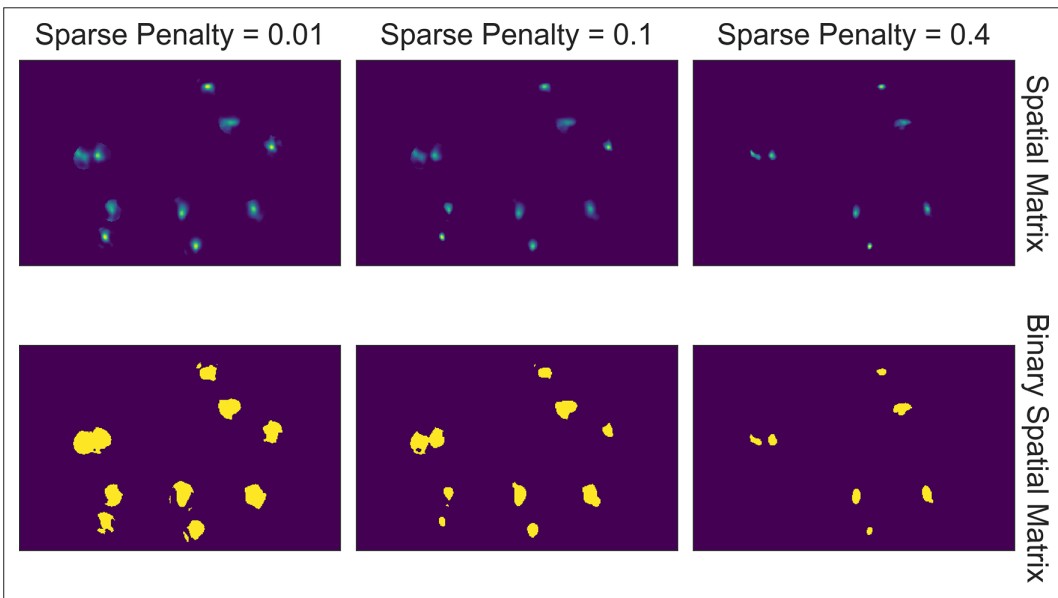

**Figure 11.** Effect of sparseness penalty in spatial update. Here, the sum projection of the spatial matrix and binarized spatial matrix is shown for three different sparse penalties. A sparseness penalty of 0.1 is considered appropriate in this case. When the sparseness penalty is set lower, artifacts begin to appear. On the other hand, when the sparseness penalty is set higher, cells are dropped out.

set much lower, many of the additional 'fragments' begin to appear in the binarized spatial footprint, even if they are not part of the cell. On the other hand, when the sparseness penalty is set too high, some cells are discarded. In the interactive visualization tool, users can inspect the temporal dynamics of these discarded cells. In general, however, we do not recommend exploiting the sparseness penalty during the spatial update to filter cells since this step does not have an explicit model of the temporal signal and thus has no power to differentiate real cells from noise.

In addition, a dilation window parameter must be specified by the user. To reduce the amount of computation when calculating how each pixel weighs onto each cell, we only update weights for cells that are close to each pixel. For each cell, an ROI is computed by performing a morphological dilation process on the previous spatial footprints of that cell. If a pixel lies outside of a cell's region of interest, this cell will not be considered when updating the pixel's weight. Thus, the dilation window parameter determines the maximum distance a cell is allowed to grow during the update compared to its previous spatial footprints. This parameter should be set large enough so that it does not interfere with the spatial update process, but at the same time not so large as to impact performance. The expected cell diameter in pixels is a good starting point.

## Temporal update

Next, we proceed to the temporal update of the CNMF algorithm. Please refer to the original paper for the detailed derivation (*Pnevmatikakis et al., 2016*). Here, given the spatial matrix and background terms, we update the temporal matrix so that it best fits the input data ($\mathbf{Y}$). First, we subtract the background term from the input data, leaving only the noisy signal from cells. We then project the data onto the spatial footprints of cells, obtaining the temporal activity for each cell. Next, we estimate a contribution of temporal activity from neighboring overlapping cells using the spatial footprints of cells and subtract it from the temporal activity of each cell. This process results in a two-dimensional matrix representing the raw temporal activity of each cell (*Friedrich et al., 2021*).

The CNMF algorithm models the relationship between the underlying 'spiking' and the calcium dynamics of a cell as an autoregressive (AR) process. It should be noted that although the underlying process that drives calcium influx is presumably cell firing, the 'spiking' signal is modeled as a continuous variable rather than a binary variable, and strictly speaking, it is only a deconvolved calcium signal. Following convention, we will refer to this variable as 'spike signal,' an approximation of the underlying cellular activity that drives calcium influx. It should be understood, however, that the exact relationship between this variable and the actual firing rate of cells is unclear since the absolute amount of fluorescence generated by a single spike, as well as the numerical effect of integrating multiple spikes on the resulting calcium signal, is unknown.

We first estimate the coefficients for the AR model. The coefficients of the AR model can be conveniently estimated from the autocorrelation of the estimated temporal activity. In addition, noise power for each cell is also estimated directly from the signal. In practice, we find that during the estimation of the AR model parameters it is helpful to first smooth the signal, otherwise the time constant of the AR model tends to be biased by high-frequency noise. Users should again use the peak-to-noise refinement cutoff frequency for both estimation of the noise power and smoothing of the signals. Finally, we update the temporal matrix by minimizing a target function for different cells, similar to what was done with the spatial matrix. Again, the target function contains a squared error term and a $\ell 1$-norm term. We also introduce a sparseness penalty parameter to control the balance between the two terms. The squared error term contains the difference between input signal and estimated calcium dynamics, while the $\ell 1$-norm term regulates the sparsity of the 'spiking' signal. Pre-estimated AR coefficients allow for a determined relationship between the 'spiking' signal and calcium dynamics for a given cell. Thus, the problem can be transformed and simplified as minimizing the target function over 'spiking' signals of different cells.

In practice, it is computationally more efficient to break down the minimization problem into smaller pieces and update subsets of cells independently and in parallel. To do so, we first identify nonoverlapping cells using a Jaccard index, which measures the amount of overlap between the spatial footprints of different cells. Once we identify these individual cells, we can update them independently so that an optimization problem and target function are formulated for each cell independently. Here, we set a cutoff Jaccard index where cells above this amount of overlap are updated in parallel. During the updating process, two additional terms are introduced: a baseline term to account for constitutive

nonzero activity of cells and an initial calcium concentration to account for a 'spiking' that started just prior to recording. The initial calcium concentration term is a scalar that is recursively multiplied by the same AR coefficient estimated for the cell. The resulting time trace, modeling the decay process of a 'spiking' event prior to the recording, is added on top of the calcium trace. The baseline activity term is also a scalar that is simply added on top of all the modeled signals. Both terms are often zero, but they are nevertheless saved and visualized. For each cell, the optimization process can be expressed formally as

$$\underset{\mathbf{c},\mathbf{b_0},c_0}{\text{minimize}} \quad \|\mathbf{yra} - \mathbf{c} - b_0 - c_0\mathbf{d}\| + \lambda\|\mathbf{Gc}\|_1$$

$$\text{subject to} \quad \mathbf{c}, \mathbf{Gc} \geq 0$$

where $\mathbf{yra}$ denotes the input movie data projected onto the spatial footprint of the given cell, $\mathbf{c}$ denotes the estimated calcium dynamic of the given cell, $b_0$ denotes the constant baseline fluorescent activity, $c_0$ denotes the initial calcium concentration, $G$ represents a matrix of AR coefficients such that $\mathbf{Gc}$ is the estimated 'spike' signal, and $\mathbf{d}$ is a vector representing the temporal decay of a single spike based on the estimated AR coefficients, such that the term $c_0\mathbf{d}$ represents the contribution of initial calcium concentration. Similar to spatial update, the scalar $\lambda$ represents the sparse penalty and controls the balance between the error term and sparsity term.

The $\ell 1$ -norm in the optimization problem is known to reduce not only the number of nonzero terms (i.e., promotes sparsity), but also the amplitude/value of nonzero terms. This effect is unwanted since in some cases the numerical of the spatial update step in CNMF algorithm and value of the resulting 'spike' signal can become too small as a side effect of promoting sparsity, making it hard to interpret and compare the 'spike' signal for downstream analysis. To counteract this phenomenon, we introduce a *post-hoc* scaling process. After the temporal update, each cell is assigned a scaling factor to

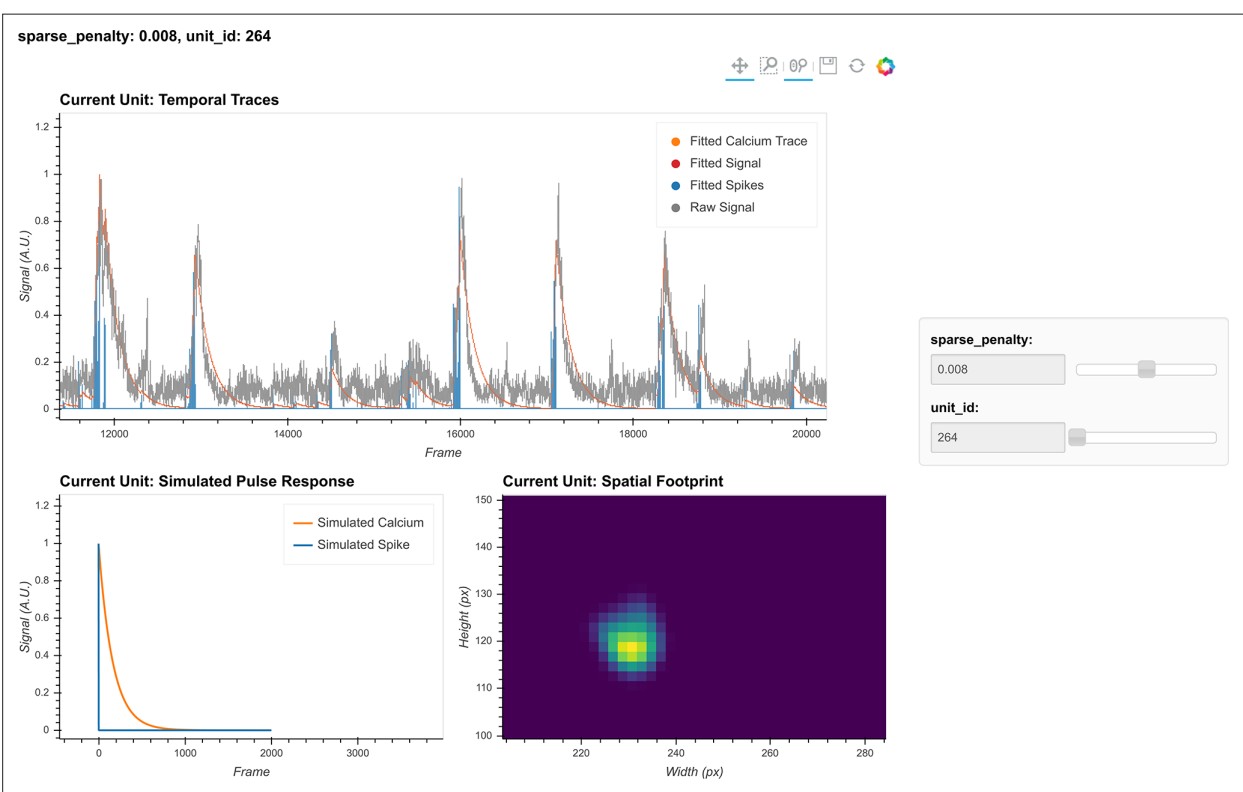

**Figure 12.** Visualization of temporal update. Here, a subset of cells is randomly chosen to pass through temporal updates with different parameters. Only one cell is visualized at a given time, and the cell can be selected using the slider on the right. The raw signal, fitted signal, fitted calcium traces, and spike signals are overlaid in the same plot. In addition, a simulated pulse response based on the estimated autoregressive parameters is plotted with the same time scale. Furthermore, the corresponding spatial footprint of the cell is plotted for cross-reference. With a given set of parameters, the user can visually inspect whether the pulse response captures the typical calcium dynamics of the cell, and whether the timing and sparsity of the spike signal fit well with the raw data. The data shown here was acquired with a frame rate of 30 fps.

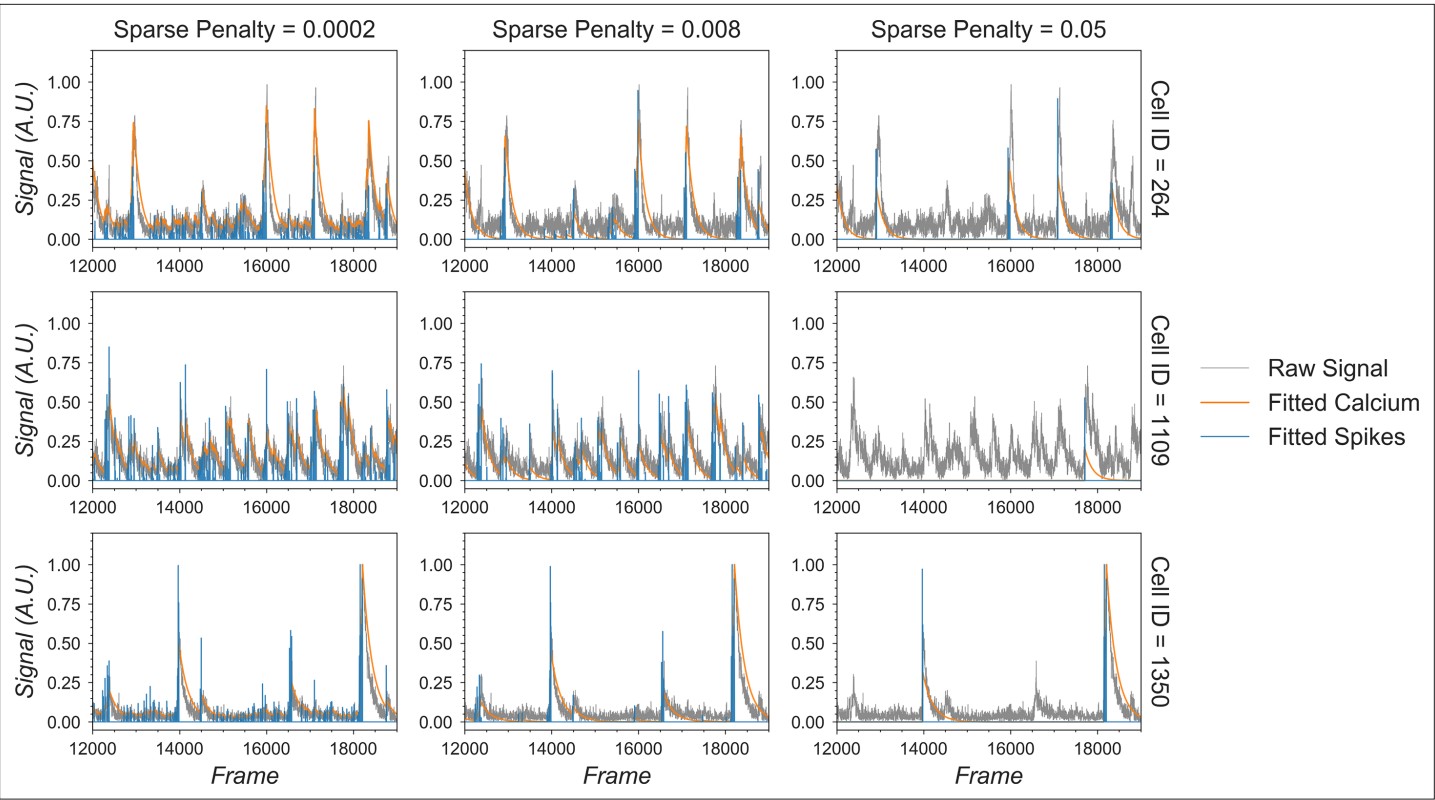

**Figure 13.** Effect of the sparseness penalty in temporal update. Here, three example cells are selected and passed to the temporal update with different sparseness penalties. The 'Raw Signal' corresponds to the input video projected onto predetermined spatial footprints. The 'Fitted Calcium' and 'Fitted Spikes' correspond to the resulting model-fitted calcium dynamics and spike signals. A sparseness penalty of 0.008 (middle column) is considered appropriate in this case. The data shown here was acquired with a frame rate of 30 fps.

scale all the fitted signals to the appropriate values. The scaling factor is solved by least-square minimizing the error between the fitted calcium signal and the projected raw signal.

The critical parameters in temporal updates are as follows: (1) the order of the AR model, usually 1 or 2. Users should choose 1 if near-instantaneous rise time is presented in the calcium dynamics of the input data (i.e., from the relatively slow sampling rate) and should choose 2 otherwise. (2) The cutoff frequency for noise used for both noise power estimation and pre-smoothing of the data during AR coefficients estimation. Users should use the values set during peak-to-noise ratio refinement. (3) The threshold for the Jaccard index determining which cells can be updated independently. Users should use a value as low as possible, as long as the speed of this step is acceptable (with large amounts of cells packed closely together, a low threshold may dramatically slow down this step), or visually inspect how sparse the spatial footprints are and determine what amount of overlap between spatial footprints results in significant crosstalk between cells. (4) The sparseness penalty is best set through visualization tools. The effect of any parameter on the temporal update can be visualized through the tool shown in *Figure 12*, where the result of the temporal update for 10 randomly selected cells is plotted as traces. There are a total of four traces shown for each cell: calcium signal, deconvolved 'spiking' signal, projected raw signal, and 'fitted signal.' The 'fitted signal' is very similar to the calcium signal and is often indistinguishable from the latter. The difference between them is that the 'fitted signal' also includes the baseline term and the initial calcium concentration term. Hence, the 'fitted signal' should better follow the projected raw signal, but it may be less interesting for downstream analysis. Toggling between different parameters triggers the dynamic update of the plots, helping the user to determine the best parameters for their data. Additionally, we highlight the effect of the sparseness penalty on resulting fitted calcium signals and spike signals in *Figure 13*. The effect is most evident in the 'fitted spikes' trace, which corresponds to the spike signal and can arguably be interpreted as a measure of the underlying neural activity per frame scaled by an unknown scalar. Here, a sparseness penalty of 0.008 is considered most appropriate. A lower sparseness penalty will introduce

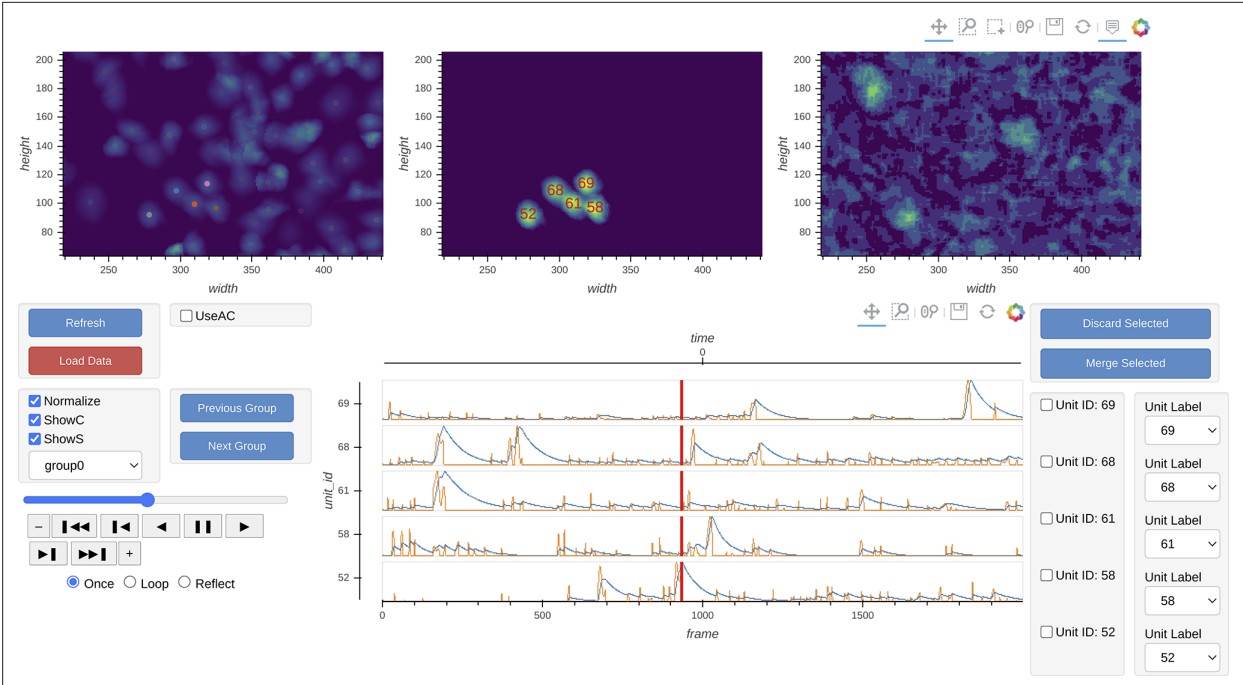

**Figure 14.** Interactive visualization of Minian output. The three images on the top show the spatial footprints of all the cells (left), spatial temporal activities of selected subset of cells (middle), and preprocessed data. The bottom row shows the display control panel (left), temporal dynamics of selected subset of cells (middle), and manual curation panel (right). The field of view, current frame, and selection of cells are all synced across different plots to help user focus on a specific region and time. The users can use the control panel to select groups of cells, change display options for temporal dynamics and spatial temporal activities, and change the current frame or play the movie. In addition, the users can directly select cells from the spatial footprints plot on the top left. The users can also directly jump to frames by double-clicking on the temporal dynamic plots. These interactive features help the users quickly focus on region and time of interests. The manual curation menu on bottom right can be used to assign unit labels to each cell, which indicate whether a cell should be dropped or merged.

many false-positive signals that do not correspond to real calcium dynamics, as can be seen in the plots. On the other hand, too high a sparseness penalty will produce false negatives where clear rises in the raw signal are not accompanied by spikes.

## Merging cells

The CNMF algorithm can sometimes misclassify a single cell as multiple cells. To counteract this phenomenon, we implement a step to merge cells based on their proximity and temporal activity. All cells with spatial footprints sharing at least one pixel are considered candidates for merging, and the pairwise correlation of their temporal activity is computed. Users can then specify a threshold where cell pairs with activity correlations above the threshold are merged. Merging is done by taking the sum of the respective spatial footprints and the mean of all of the temporal traces for all cells to be merged. Since this is only a simple way to correct for the number of estimated cells and does not fit numerically with what the model CNMF assumes, merging is only done between iterations of CNMF, but not at the end.

## Manual curation

Minian provides an interactive visualization to help the users manually inspect the quality of putative cells and potentially merge or drop cells. At any given time, the visualization shows spatial temporal activities (*Figure 14*, top row, middle panel) and temporal dynamics of a selected subset of cells (*Figure 14*, bottom row ). The spatial temporal activities are shown side-by-side with the spatial footprints of all cells and the preprocessed movie (input to CNMF algorithm) at a given frame (*Figure 14*, top row). The field of view is synchronized across the three images on the top, so that the users can easily zoom in and compare the estimated spatial footprints of cells to the input data. The spatial temporal images in the middle show the product of spatial footprints and calcium dynamics, which

represent the model estimated image of a subset of cells at a given frame. This spatial temporal product is calculated on-the-fly and synchronized with the frame indicators on the temporal dynamic plots. In this way, users can easily pick times of interest (e.g., when a cell has a calcium event) and validate whether the estimated spatial temporal activities match the input data. Lastly, this interactive visualization allows the user to either drop false-positive cells or merge multiple cells together via drop-down menus. The result of manual curation is saved as an array with a label for each unit indicating whether a cell should be discarded or how several cells should be merged. In this way, only the new label is saved and no data is modified, allowing the user to repeat or correct the manual curation process if needed.

## Cross-registration

After completing the analysis of individual recording sessions, users can register cells across sessions. While more complex approaches are proposed in other pipelines (*Giovannucci et al., 2019*; *Sheintuch et al., 2017*), here, our intention is simplicity. To account for shifts in the field of view from one session to the next, we first align the field of view from each session based upon a summary frame. Users can either choose a max projection of each preprocessed and motion-corrected video, or a summed projection of the spatial footprints of all cells. Users can also choose which session should be used as the template for registration, to which every other session should be aligned. We use a standard cross-correlation based on a template-matching algorithm to estimate the translational shifts for each session relative to the template and then correct for this shift. The weighted centroid of each cell's spatial footprint is then calculated and pairwise centroid distances are used to cross-register cells. A distance threshold (maximum pixel distance) is set. Users should choose this threshold carefully to reflect the maximum expected displacement of cells across sessions after registration. We found that a threshold of five pixels works well. Finally, a pair of cells must be the closest cells to each other in order to be considered the same cell across sessions.

To extend this method to more than two sessions, we first cross-register all possible session pairs. We then take the union of all these pairwise results and transitively extend the cross-registration across more than two sessions. At the same time, we discard all matches that result in conflicts. For example, if cell A in the first session is matched with cell B in the second session, and cell B is in turn matched with cell C in the third session, but cells A and C are not matched when directly registering the first and third sessions, all of these matches are discarded and all three cells are treated as individual cells. We recognize that this approach might be overly conservative. However, we believe that this strategy provides an easy-to-interpret result that does not require users to make decisions about whether to accept cell pairs that could conflict across sessions.

To save computation time, we implement a moving window where centroid distances are only calculated for cell pairs within these windows. Users should set the size of windows to be much larger than the expected size of cells.

## Hardware and dependencies

Minian has been tested using OSX, Linux, and Windows operating systems. Additionally, although we routinely use Minian on specialized analysis computers, the pipeline works on personal laptops for many common length (~30 min) miniature microscope experiments. Specifications of all of the computers that have been tested can be found in Tested hardware specifications. We anticipate that any computer with at least 16 GB of memory will be capable of processing at least 20 min of recording data, although increased memory and CPU power will speed up processing. Moreover, due to the read-write processes involved in out-of-core computation, we recommend that the videos to be processed are held locally at the time of analysis, preferably on a solid-state drive. The relatively slow speed of transfer via Ethernet cables, Wi-Fi, or USB cables to external drives will severely impair analysis times.

Minian is built on top of project Jupyter (*Kluyver et al., 2016*) and depends heavily on packages provided by the open-source community, including NumPy (*Harris et al., 2020*), SciPy (*Virtanen et al., 2020*), xarray (*Hoyer and Hamman, 2017*), HoloViews (*Rudiger et al., 2020*), Bokeh (*Bokeh Development Team, 2020*), OpenCV (*Bradski, 2020*), and Dask (*Dask Team, 2016*). A complete list of direct dependencies for Minian can be found in 'List of dependencies.' Of note, the provided install instructions handle the installation of all dependencies.

## Results

To validate the accuracy as well as benchmark the performance of Minian, we ran the Minian pipeline on a series of simulated and experimental datasets and compare the output and performance to those obtained with CaImAn, which is one of the most widely adapted calcium imaging analysis pipelines in the field. In addition, we also validated the full workflow of Minian by applying the pipeline to several recordings of animals running on a linear track and looked at the stability of place cells. These results are presented in the following sections.

### Validation with simulated datasets

We first validated Minian with simulated datasets. We synthesized different datasets with varying number of cells and signal levels based on existing works (*Zhou et al., 2018*; *Lu et al., 2018*). The simulated datasets contain local background fluctuations, noise, and motions similar to experimental datasets (see 'Generation of simulated datasets' for details). The field of view contains 512 × 512 pixels and 20,000 frames, corresponding to roughly 10 min of recording at 30 fps. We processed the data with both Minian and CaImAn. For Minian, we utilized the visualization described here to optimize the parameters. For CaImAn, we used the same parameters as Minian whenever the implementations were equivalent. Otherwise, we followed the suggested parameters and tweaked them based on the knowledge of simulated ground truth.

To compare the results objectively, we first matched the resulting putative cells from the output of Minian or CaImAn to the simulated ground truth (see 'Matching neurons for validation' for details). We then calculated three metrics to measure the quality of output: F1 score, spatial footprints correlation, and temporal dynamics correlation. The F1 score is defined as the harmonic mean of precision (proportion of detected neurons that are true) and recall (proportion of ground-truth neurons that has been detected). Hence, the F1 score measures the overall accuracy of neuron detection. For each detected neuron that has been matched to ground truth, we compute Pearson correlation between the estimated and ground-truth spatial footprint, as well as the Pearson correlation between the estimated calcium dynamic and the ground-truth calcium dynamic. We then take the median correlation across all the matched neurons to measure the overall quality of estimated spatial footprints and temporal dynamics.

As shown in *Figure 15*, both Minian and CaImAn achieve similar and near-perfect levels (>0.95) of F1 score across all conditions. Similarly, the spatial footprints remain nearly perfect (>0.95) for both pipelines across all conditions. At the lowest signal level (0.2), both pipelines suffer from decreased correlation of temporal dynamics. This is likely due to noise and background contaminating the true signal. Overall, these results show that the Minian and CaImAn pipelines perform similarly well in terms of output accuracy on simulated datasets.

Additionally, we want to validate the deconvolved signal from Minian output since this is usually the most important output for downstream analysis. Our ground-truth spikes are simulated as binary signals. However, in reality calcium activity often reflects the integration of several spikes, and the deconvolved signals from Minian output are real-valued. Because of this, we downsampled both the ground-truth spikes and deconvolved signals by five times, and then calculated Pearson correlation for all matched cells. The resulting correlation is summarized in *Figure 16A*. Our results indicate that the deconvolved output from Minian is highly similar to ground-truth spikes when signal level is high, and the correlation asymptote and approach 1 when signal level is higher than 1. The lower correlation corresponding to low signal level is likely due to the background and noise contamination being stronger than signal. In line with this idea, the detected 'spikes' from the deconvolved signals closely match those from ground truth, as shown by the example traces in *Figure 16B*. The main difference between the two traces is the amplitude of the deconvolved signals, which is prone to be influenced by local background and noise. Overall, these results suggest that Minian can produce deconvolved signals that are faithful to ground truth and suitable for downstream analysis.

### Validation with experimental datasets

We next validated Minian with experimental datasets. The data was collected from hippocampal CA1 regions in animals performing a spatial navigation task. Six animals with different density of cells were included in the validation dataset. The recordings are collected with 608 × 608 pixels at 30 fps and last 20 min (~36,000 frames). Due to difficulties in obtaining ground truth for experimental data,

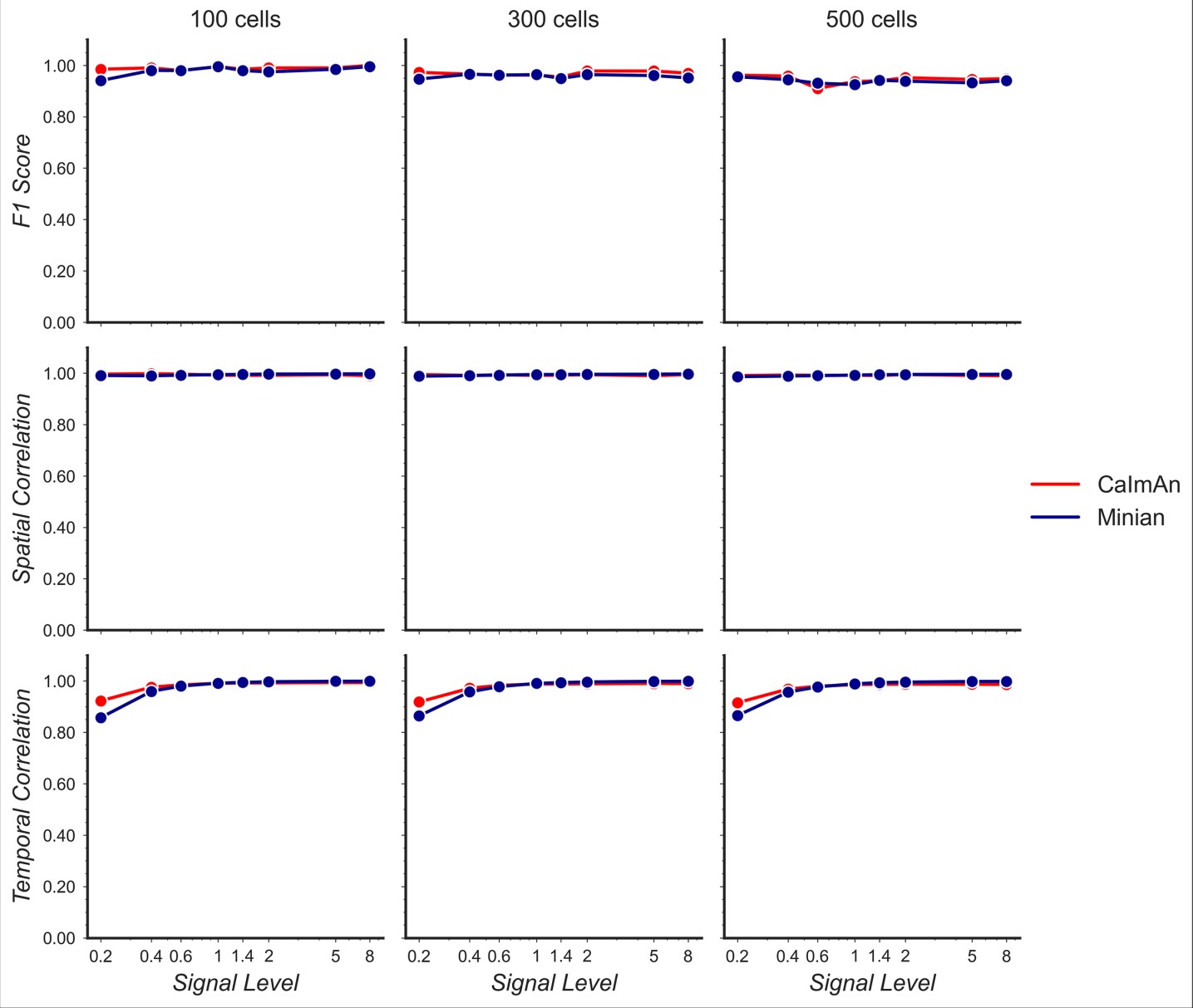

**Figure 15.** Validation of Minian with simulated datasets. Simulated datasets with varying signal level and number of cells are processed through Minian and CaImAn. The F1 score (top), median correlation of spatial footprints (middle), and median correlation of temporal dynamics (bottom) are plotted as a function of signal level. Both pipelines achieve near-perfect (>0.95) F1 scores and spatial footprint correlation across all conditions. The correlation of temporal dynamics is lower when the signal level is 0.2, but remains similar across the two pipelines overall.

The online version of this article includes the following source data for figure 15:

**Source data 1.** Raw validation performance with simulated data.

we choose to validate Minian with CaImAn, which has been established as one of the most accurate existing pipelines. To evaluate the results objectively, we matched resulting ROIs from Minian with those from CaImAn using the same approach as in 'Validation with simulated datasets.' We then calculated correlation of spatial footprints and temporal activity between matched ROIs from the two pipelines. Across the six datasets, the mean F1 score is 0.73 (sem ± 0.03). The mean spatial footprints correlation is 0.84 (sem ± 0.02), and the mean temporal activity correlation is 0.86 (sem ± 0.02). An example field of view and temporal activity from matched ROIs is shown in **Figure 17**. Our results indicate that most of the ROIs detected by Minian and CaImAn correspond to the same population of putative cells, and the resulting spatial footprints and temporal activity are nearly identical. These

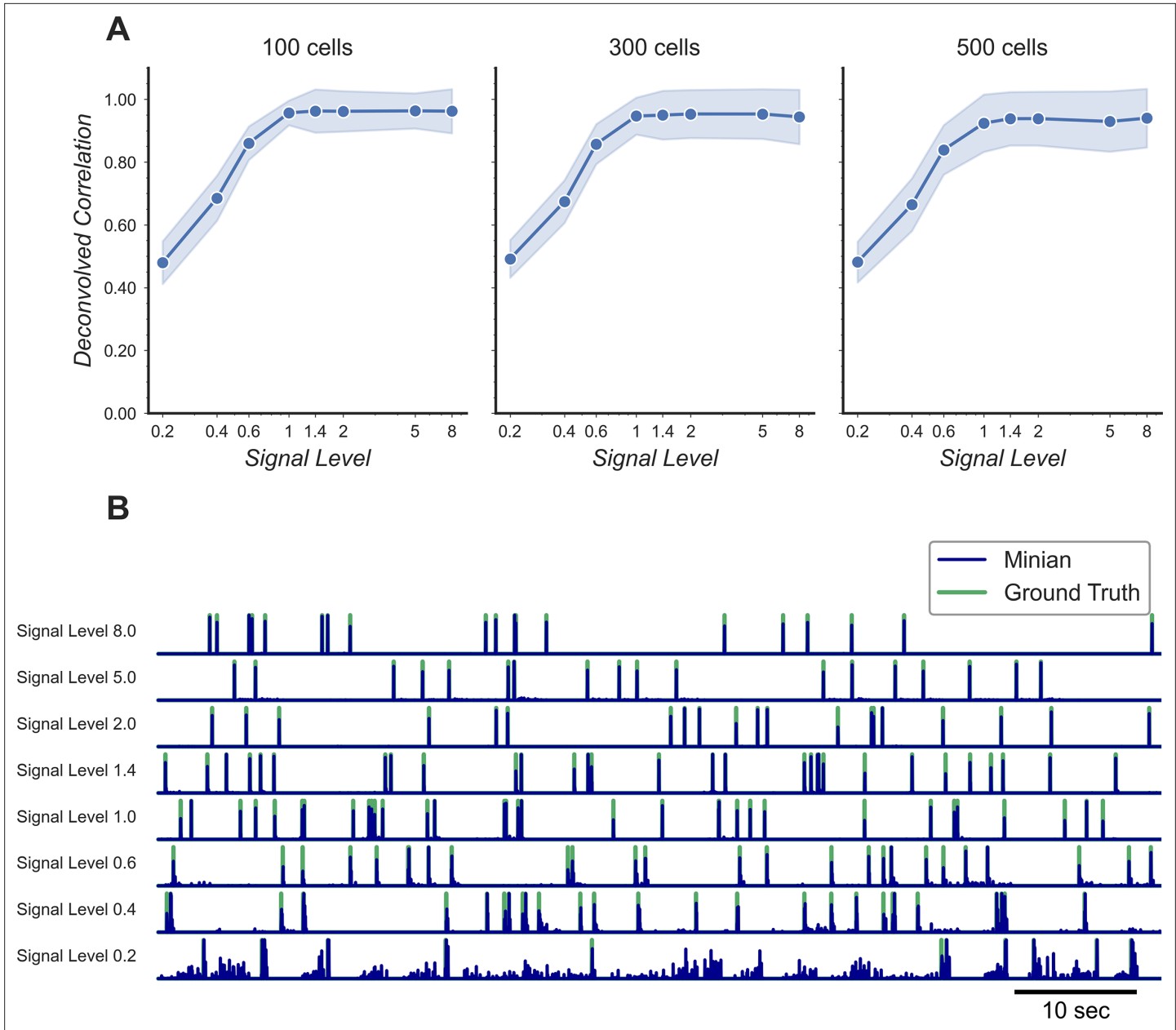

**Figure 16.** Validation of deconvolved signal from Minian. (**A**) Correlation of deconvolved signals from Minian output with simulated ground truth. The mean correlation across all cells (blue line) and standard deviation (light blue shade) are shown separately for different signal levels and number of cells. The correlation asymptote and approach 1 when signal level is higher than 1. (**B**) Example deconvolved traces from Minian output overlaid with simulated ground truth. One representative cell is drawn from each signal level. The binary-simulated spikes are shown in green, with the real-valued Minian deconvolved output overlaid on top in blue. The deconvolved signals closely match the ground truth, and the main difference between the two signals is in the amplitude of the deconvolved signals, which tend to be influenced by local background.

The online version of this article includes the following source data for figure 16:

**Source data 1.** Raw correlations between Minian deconvolved traces and simulated ground truth.

**Source data 2.** Raw example traces from Minian and simulated ground truth.

cells tend to cluster near the center of the field of view, which usually has better signal-to-noise ratio. However, the cells near the edge of the field of view usually have low intensity and spatial consistency due to the optical property of GRIN lens. As a consequence, Minian and CalmAn might detect different populations of cells near the border of field of view due to differences in preprocessing and initialization between the two pipelines. We have chosen to use the same set of parameters across

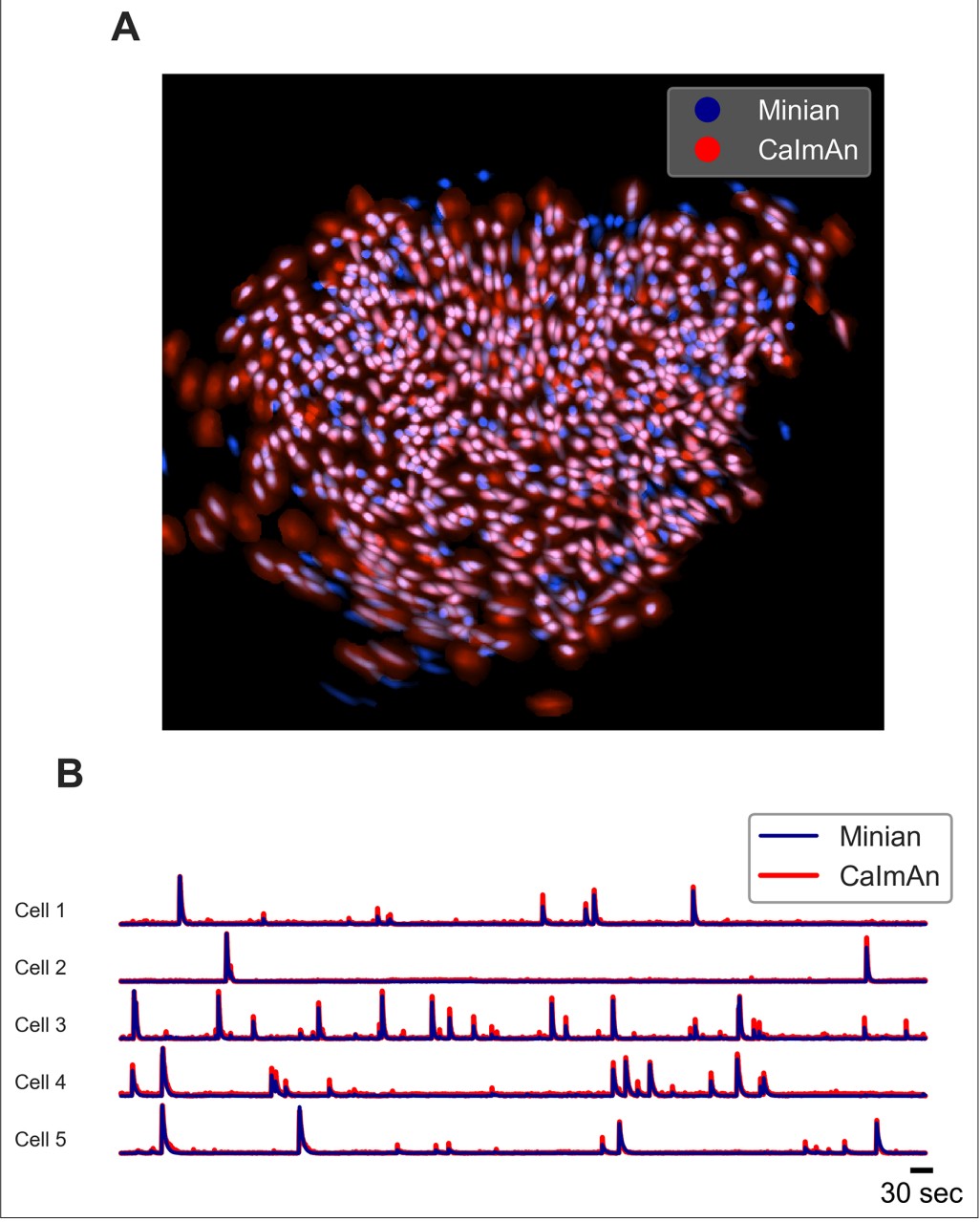

**Figure 17.** Example output of Minian and CalmAn with experimental datasets. (**A**) An example field of view from one of the experimental datasets. The spatial footprints from Minian and CalmAn are colored as blue and red, respectively, and overlaid on top of each other. Most of the spatial footprints from both pipelines overlap with each other. (**B**) Five example matched temporal activity from Minian and CalmAn overlaid on top of each other. The extracted temporal activity is highly similar across the two pipelines.

The online version of this article includes the following source data for figure 17:

**Source data 1.** Raw spatial footprint values shown in the overlay plot.

**Source data 2.** Raw example traces from Minian and Caiman.

all datasets so that the results are easier to interpret, hence the parameters we used were relatively conservative. In practice, the users can further fine-tune the parameters for each recording so that Minian would be able to capture all the low signal cells in the field of view. Overall, these results suggest that the output of Minian is highly similar to CalmAn when analyzing experimental datasets.

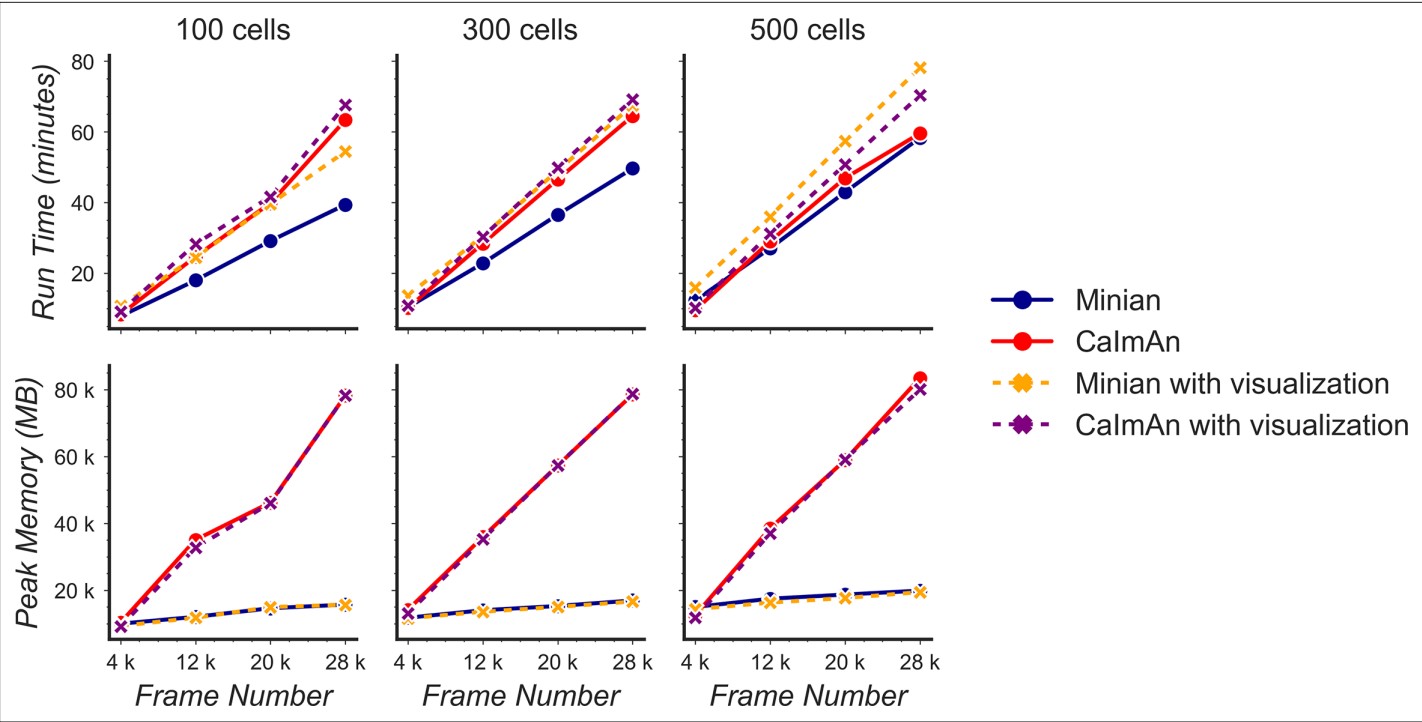

**Figure 18.** Benchmarking of computational performance. Data with varying number of cells and frames were processed through Minian and CaImAn. The run time (top) and peak memory usage (bottom) were recorded and plotted as a function of frame number. For both pipelines, the run time scales linearly as a function of the number of frames and remains similar across the pipelines. However, the peak memory usage for CaImAn also scales linearly as the number of frames increases, while Minian maintains a relatively constant peak memory usage across different frame numbers and cell numbers.

The online version of this article includes the following source data for figure 18:

**Source data 1.** Raw memory usage and running time with different datasets for both pipelines.

## Benchmarking computational performance

To see how the performance of Minian scales with different input data size, we synthesized datasets with varying number of cells and number of frames (recording length). The field of view contains 512 × 512 pixels (same as those used in validation of accuracy), and the signal level was held constant at 1 to make sure both Minian and CaImAn can detect roughly equal number of neurons during the pipeline. To this end, we tracked two metrics of performance: the total running time of the pipeline and the peak memory usage during running. The running time was obtained by querying operating system time during the pipeline. The memory usage was tracked with an independent process that queries memory usage of the pipeline from the operating system on a 0.5 s interval. Both pipelines were set to utilize four parallel processes during the run across all conditions. All benchmarking are carried out on a custom-built Linux machine (model 'Carbon' under Tested hardware specifications).

As shown in *Figure 18*, the run time of both Minian and CaImAn scales linearly as a function of input recording length. The exact running times vary depending on the number of cells as well as whether visualization is included in the processing, but in general the running time is similar across both pipelines. On the other hand, the peak memory usage of CaImAn scales linearly with recording length when the number of parallel processes was set to be constant. At the same time, the peak memory usage of Minian stays mostly constant across increasing number of frames. This is likely due to the flexible chunking implementation of Minian (see 'Parallel and out-of-core computation with Dask'), where Minian was able to break down computations into chunks in both spatial and temporal dimensions depending on which way is more efficient. In contrast, CaImAn only splits data into different spatial chunks (patches), resulting in a linear scaling of memory usage with recording length for each chunk-wise computation. Additionally, we run Minian and CaImAn with different number of parallel processes on the simulated dataset with 28,000 frames and 500 cells. As expected, with more parallel processes the performance improves and the run time decreases but at the same time the

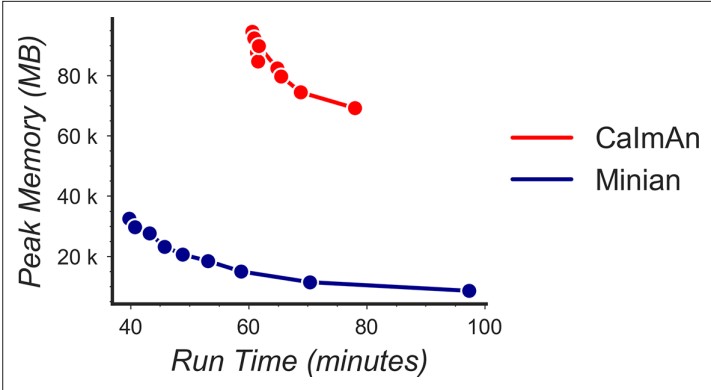

**Figure 19.** Tradeoff between run time and memory usage. Simulated data with 500 cells and 28,000 frames were processed through Minian and CaImAn with different numbers of parallel processes. We varied the number of parallel processes from 2 to 10, and the resulting memory usage is plotted as a function of run time. For both pipelines, the curve takes a hyperbola shape, showing the tradeoff between run time and memory usage.

The online version of this article includes the following source data for figure 19:

**Source data 1.** Raw memory usage and running time with different parallel processes for both pipelines.

---

total peak memory usage increases. The tradeoff between run time and peak memory usage is shown in *Figure 19*. In conclusion, these results show that in practice Minian is able to perform as fast as CaImAn, while maintaining near constant memory usage regardless of input data size. This allows the users to process much longer recordings with limited RAM resources.

## Validation with hippocampal CA1 place cells

In addition to direct validation of the output for single session, we wanted to validate the scientific significance of the spike signal, as well as the quality of the cross-session registration, and ensure that Minian is capable of generating meaningful results consistent with the existing literature. We leveraged the extensively documented properties of place cells in rodent hippocampal CA1 (*O'Keefe and Dostrovsky, 1971*). Place cells have been shown to have consistent place fields across at least 2 days (*Ziv et al., 2013*; *Thompson and Best, 1990*) with only a minority of detected cells undergoing place field remapping. Here, we looked at place field stability across two linear track sessions (*Figure 20A*). Briefly, animals were trained to run back and forth on a 2 m linear track while wearing a Miniscope to obtain water rewards available at either end (*Shuman et al., 2020*). The time gap between each session was 2 days. We record calcium activity in dorsal CA1 region with a FOV of 480 × 752 pixels collected at 30 fps. Each recording session lasts 15 min (~27,000 frames). Calcium imaging data were analyzed with Minian, while the location of animals was extracted with an open-source behavioral analysis pipeline ezTrack (*Pennington et al., 2019*). The resulting calcium dynamics and animal behavior were aligned with the timestamps recorded by Miniscope data acquisition software (http://miniscope.org/index.php/Main_Page). We used the spike signal for our downstream analysis. To calculate average spatial activity rate, we binned the 2-m-long track into 100 spatial bins. In addition, we separated the epochs when the animals are running in opposite directions, resulting in a total of 200 spatial bins. We then smoothed both the binned activity rate and animal's occupancy with a Gaussian kernel with a standard deviation of 5 cm. We classified place cells based on three criteria: spatial information criterion, stability criterion, and place field size criterion (*Shuman et al., 2020*). (See 'Classification of place cells' for more detail.) Finally, we analyzed cells that are cross-registered by Minian and are classified as place cells in both sessions. We then calculated the Pearson correlation for the average spatial firing rate for each cross-registered cell. We found that, on average, place cells have a correlation of ~0.6, which is consistent with the existing literature (*Shuman et al., 2020*).

Next, we validated the cross-session registration to verify that the correct cells were being matched across days. We translated the spatial footprints of the second session in both directions up to 50 pixels and registered the cells with the shifted spatial footprints. We then carried out the same analysis with the registration results from shifted spatial footprints. We found that the average correlations between spatial firing patterns have higher values when the shifts are close to zero (*Figure 20B*).

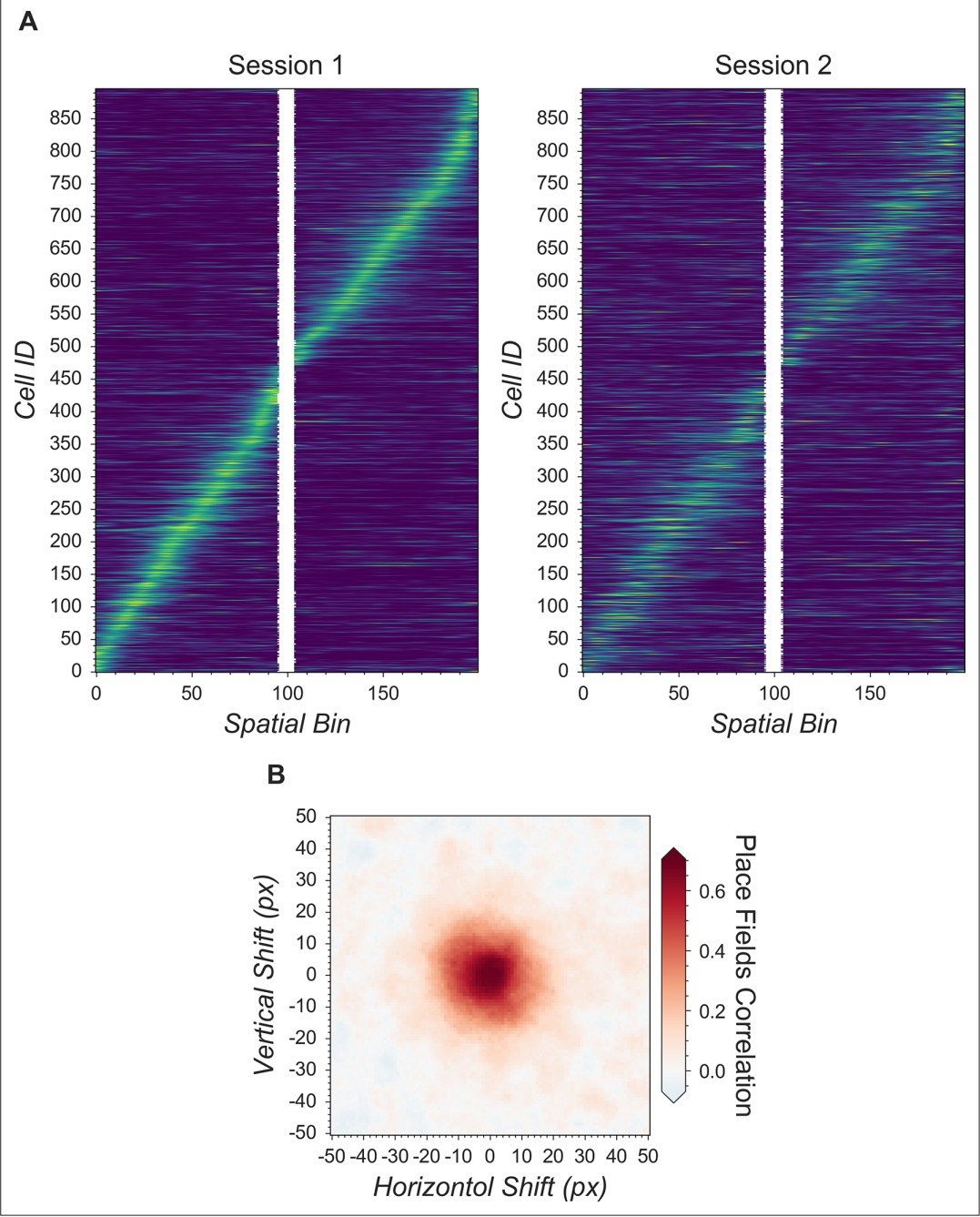

**Figure 20.** Validation of Minian with hippocampal CA1 place cells. (**A**) Matching place cells from two recording sessions. The cells are matched from one session to the other using the cross-session registration algorithm and sorted based on place field in the first session. In both sessions, animals run on a 2-m-long linear track with water reward at both ends. The track is divided into 200 spatial bins. The mean 'firing' rate calculated from the spike signal for each cell is shown. Cell IDs are assigned by Minian when each session is analyzed independently. (**B**) Averaged correlations of spatial firing rates with different artificial shifts. We artificially shifted the spatial footprints of the second linear track session, then carried out registration and calculated a mean correlation of spatial firing rates for all place cells. The artificial shifts were relative to the aligned spatial footprints and range from –50 to 50 pixels.

The online version of this article includes the following source data for figure 20:

**Source data 1.** Raw correlation of spatial firing pattern with different shifts in field of view.

**Source data 2.** Raw spatial firing activity for the two sessions shown.

In conclusion, Minian can reliably process in vivo calcium imaging data and produce results that are in agreement with the known properties of rodent CA1. Minian can thus help neuroscience labs easily implement and select the best parameters for their calcium analysis pipeline by providing detailed instructions and visualizations.

## Discussion

### Making open science more accessible

Neuroscience has benefited tremendously from open-source projects, ranging from do-it-yourself hardware (*White et al., 2019*) to sophisticated algorithms (*Freeman, 2015*). Open-source projects are impactful because they make cutting-edge technologies available to neuroscience labs with limited resources, as well as opening the door for innovation on top of previously established methods. We believe that openly sharing knowledge and tools is just the first step. Making knowledge accessible even to nonexperts should be one of the ultimate goals of open-source projects.

With the increasing popularity of miniaturized microscopes (*Aharoni and Hoogland, 2019*), there has been significant interest in analysis pipelines that can reliably extract neural activities from the data. Numerous algorithms have been developed to solve this problem (*Zhou et al., 2018*; *Mukamel et al., 2009*; *Klibisz et al., 2017*; *Friedrich et al., 2017*; *Giovannucci et al., 2017*; *Pnevmatikakis et al., 2016*), and many of them are implemented as open-source packages that can function as a one-stop pipeline (*Lu et al., 2018*; *Jun et al., 2017*; *Giovannucci et al., 2019*). However, one of the biggest obstacles for neuroscience labs in adopting analysis pipelines is the difficulty in understanding the exact operation of the algorithms, leading to two notable challenges: first, researchers face difficulties adjusting the parameters when the data they have collected are out of the expected scope of the pipeline's default parameters. Second, even after neural activity data is obtained, it is hard for researchers to be sure that they have chosen the best approaches and parameters for their dataset. Indeed, it has been found that depending on the features of the data and the metric used, more sophisticated algorithms do not always outperform simpler algorithms (*Pachitariu et al., 2018*), making it even harder for researchers to interpret the results obtained from some analysis pipelines. Researchers therefore often have to outsource data analysis to experts with strong computational backgrounds or simply trust the output of the algorithms being used. Minian was created to address these challenges. By providing not only detailed documentation of all functions, but also by providing rich interactive visualizations, Minian helps researchers to develop an intuitive understanding of the operations of algorithms without expertise in mathematics or computer science. These insights help researchers choose the best parameters, as well as to become more confident in their interpretation of results. Furthermore, transparency regarding the underlying algorithms enables researchers to develop in-house modifications of the pipeline, which is a common practice in neuroscience labs. We believe that Minian will contribute to the open science community by making the analysis of calcium imaging data more accessible and understandable to neuroscience labs.

### Limitations

Although Minian provides users with insights into the parameter tuning process across different brain regions, these insights are achieved mainly through visual inspection. However, the performance of an analysis pipeline should be measured objectively. While calcium imaging has been validated with electrophysiology under ex vivo settings (*Chen et al., 2013*), ground-truth data for single-photon in vivo calcium imaging are lacking, making objective evaluation of the algorithms difficult. Therefore, here we have provided only indirect validations of the pipeline by recapitulating well-established biological findings.

## Acknowledgements

We thank Eftychios A Pnevmatikakis, Andrea Giovannucci, and Liam Paninski for establishing the theoretical foundation and providing helpful insights for the pipeline. We thank Pat Gunn for helping with benchmarking with CalmAn pipeline. We thank Taylor Francisco and Denisse Morales-Rodriguez for helping with data analysis and revision. We thank MetaCell (Stephen Larson, Giovanni Idili, Zoran Sinnema, Dan Knudsen, and Paolo Bazzigaluppi) for contributing to the documentation and continuous integration of the pipeline. We thank Stellate Communications for assistance with the

preparation of this manuscript. We thank Brandon Wei, Mimi La-Vu, and Christopher Lee for contributing to the dataset used in Minian development and testing. The authors acknowledge support from the following funding sources: WM is supported by NIH F32AG067640. KR is supported by NIH BRAIN Initiative (R01EB028166), James S. McDonnell Foundation's Understanding Human Cognition Scholar Award (220020466), NSF Award (NSF1926800 and NSF2046583), Simons Foundation (891834) and Alfred P. Sloan Foundation (FG-2019-12027). TS is supported by CURE Epilepsy Taking Flight Award, American Epilepsy Society Junior investigator Award, R03 NS111493, R21 DA049568, and R01 NS116357. DA is supported by NIH U01NS094286-01, and NSF Award (NSF1700408 Neurotech Hub). DJC is supported by NIH DP2MH122399, R01 MH120162, One Mind Otsuka Rising Star Award, McKnight Memory and Cognitive Disorders Award, Klingenstein-Simons Fellowship Award in Neuroscience, Mount Sinai Distinguished Scholar Award, Brain Research Foundation Award, and NARSAD Young Investigator Award.

## Additional information

### Competing interests

Denise J Cai: Reviewing editor, *eLife*. The other authors declare that no competing interests exist.

### Funding

| Funder | Grant reference number | Author |
| --- | --- | --- |
| National Institutes of Health | DP2MH122399-01 | Denise J Cai |
| National Institutes of Health | R01MH120162-01A1 | Denise J Cai |
| One Mind | Otsuka Rising Star Award | Denise J Cai |
| McKnight Foundation | | Denise J Cai |
| Klingenstein-Simons Fellowship | | Denise J Cai |
| Icahn School of Medicine at Mount Sinai | Distinguished Scholar Award | Denise J Cai |
| Brain Research Foundation | | Denise J Cai |
| NARSAD Young Investigator Award | | Denise J Cai |
| NIH | R01EB028166 | Kanaka Rajan |
| James S. McDonnell Foundation | 220020466 | Kanaka Rajan |
| National Science Foundation | 1926800 | Kanaka Rajan |
| National Science Foundation | 2046583 | Kanaka Rajan |
| Simons Foundation | 891834 | Kanaka Rajan |
| Alfred P. Sloan Foundation | FG-2019-12027 | Kanaka Rajan |
| NIH | U01NS094286-01 | Daniel Aharoni |
| National Science Foundation | 1700408 Neurotech Hub | Daniel Aharoni |
| National Institutes of Health | F32AG067640 | William Mau |
| CURE | Epilepsy Taking Flight Award | Tristan Shuman |

| Funder | Grant reference number | Author |
| --- | --- | --- |
| American Epilepsy Society | Junior investigator Award | Tristan Shuman |
| National Institutes of Health | R03 NS111493 | Tristan Shuman |
| National Institutes of Health | R21 DA049568 | Tristan Shuman |
| National Institutes of Health | R01 NS116357 | Tristan Shuman |

The funders had no role in study design, data collection and interpretation, or the decision to submit the work for publication.

### Author contributions

Zhe Dong, Conceptualization, Formal analysis, Investigation, Software, Validation, Visualization, Writing - original draft; William Mau, Conceptualization, Investigation, Project administration, Resources, Validation, Writing - review and editing; Yu Feng, Data curation, Resources, Validation, Writing - review and editing; Zachary T Pennington, Conceptualization, Investigation, Resources, Validation, Writing - review and editing; Lingxuan Chen, Yosif Zaki, Resources, Validation, Writing - review and editing; Kanaka Rajan, Conceptualization, Formal analysis, Resources, Writing - review and editing; Tristan Shuman, Conceptualization, Data curation, Resources, Supervision, Writing - review and editing; Daniel Aharoni, Conceptualization, Formal analysis, Investigation, Resources, Supervision, Writing - review and editing; Denise J Cai, Conceptualization, Funding acquisition, Project administration, Resources, Supervision, Writing - review and editing

### Author ORCIDs

Zhe Dong http://orcid.org/0000-0002-7366-8939
William Mau http://orcid.org/0000-0002-3233-3243
Tristan Shuman http://orcid.org/0000-0003-2310-6142
Daniel Aharoni http://orcid.org/0000-0003-4931-8514
Denise J Cai http://orcid.org/0000-0002-7729-0523

### Ethics

All experiments were performed in accordance with relevant guidelines and regulations approved by the Institutional Animal Care and Use Committee of Icahn School of Medicine at Mount Sinai (Reference #: IACUC-2017-0361, Protocol #: 17-1994).

### Decision letter and Author response

Decision letter https://doi.org/10.7554/eLife.70661.sa1
Author response https://doi.org/10.7554/eLife.70661.sa2

## Additional files

### Supplementary files

• Transparent reporting form

### Data availability

Github repo: https://github.com/denisecailab/minian (copy archived at swh:1:rev:a6d3339932df3c63aa46811fc70fc00aca09218d).

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

# Appendix 1

## Supplemental information

### Parallel and out-of-core computation with Dask

In Minian, we use a modern parallel computing library called Dask to implement parallel and out-of-core computation. Dask divides the data into small chunks along all dimensions, then flexibly merges the data along some dimensions in each step. We leverage the fact that each step in our pipeline can be carried out chunk by chunk independently along either the temporal (frame) dimension or the spatial (height and width) dimensions, thus requiring no interpolation or special handling of borders when merged together, producing results as if no chunking had been done. For example, motion correction and most preprocessing steps that involve frame-wise filtering can be carried out on independent temporal chunks, whereas computation of pixel correlations can be carried out on independent spatial chunks. Similarly, during the core CNMF computation steps, spatial chunking can be used during update of spatial footprints since spatial update is carried out pixel by pixel. Meanwhile, temporal chunking can be used when projecting the input data onto spatial footprints of cells, which is usually the most memory-demanding step. Although the optimization step during the temporal update is computed across all frames and no temporal chunking can be used, we can still chunk across cells, and in practice the memory demand in this step is much smaller compared to other steps involving raw input data. Consequently, our pipeline fully supports out-of-core computation, and memory demand is dramatically reduced. In practice, a modern laptop can easily handle the analysis of a full experiment with a typical recording length of up to 20 min. Dask also enables us to carry out lazy evaluation of many steps where the computation is postponed until the result is needed, for example, when a plot of the result is requested. This enables selective evaluation of operations only on the subset of data that will become part of the visualization and thus helps users to quickly explore a large space of parameters without committing to the full operation each time.

### Seeds refinement with a Gaussian mixture model

As described earlier, an alternative strategy to thresholding fluorescence intensity during seeds initialization is to explicitly model the distribution of fluorescence fluctuations of all candidate seeds and select those with relatively higher fluctuation. Here, we describe this process and the rationale. Since the seeds are generated from local maxima, they include noise from relatively empty regions with no actual cells. The seeds from these regions usually have low fluctuations in fluorescence across time and can be classified as spurious. To identify these cases, we compute a range of fluctuation for each seed (range of min-max across time) and model these ranges with a Gaussian mixture model of two components. The fluctuations from 'noise' seeds compose a Gaussian distribution with low fluctuation, while seeds from actual cells assume a higher degree of fluctuation and form another Gaussian distribution with a higher mean. Any seed whose fluctuations belong to the lower Gaussian distribution is discarded in this step. To compute the range of fluctuation for each seed, we compute the difference between the 99.9 and 0.1 percentile of all fluorescence values across time, which is less biased by outliers than the actual maximum and minimum values.

Normally, this step is parameter-free. In rare cases, there are regions containing noise while other regions are almost completely dark. Thus, seeds from these two regions will form two peaks in the distribution of what the user would consider 'bad seeds,' and a Gaussian mixture model with two components will no longer be valid. In such cases, users can tweak the number of components (number of modeled Gaussian distributions), as well as the number of components to be considered as composed of real signal. However, because the two noise distributions are likely to overlap to some degree, using two components will likely suffice. The distribution of fluctuations, Gaussian mixture model fit, and resulting seeds are visualized, enabling the user to judge the appropriateness and accuracy of this step. It should be noted that in practice we have found this process to depend heavily on the relative proportion of the 'good' and 'bad' seeds and can easily result in a significant amount of false negatives if the proportion of the 'bad' seed is too low. This makes the Gaussian mixture model approach less stable and in general less preferable to simple thresholding unless a good threshold of fluorescence intensity cannot be easily determined.

### Generation of simulated datasets

We use a pipeline modified from (*Zhou et al., 2018*) and (*Lu et al., 2018*) to generate simulated data for validation and benchmarking of Minian. Specifically, we generate a 512 × 512 pixels field of view with varying number of frames and neurons. The neurons are simulated as spherical 2-D Gaussian.

The center of neurons is drawn uniformly from the whole field of view, and the Gaussian widths $\sigma_x$ and $\sigma_y$ for each neuron are drawn from $\mathcal{N}(15, 5^2)$, with a minimum value of 3. Spikes are simulated from a Bernoulli process with a 0.01 probability of spiking per frame. Calcium dynamics are simulated by convolving the spikes with a temporal kernel $g(t) = exp(-t/\tau_d) - exp(-t/\tau_r)$, with rise time $\tau_r = 5$ frames and decay time $\tau_d = 60$ frames. We simulate the spatial footprints of backgrounds as spherical 2-D Gaussian distributed uniformly across field of view. In total, 300 independent background terms are used for all simulations. The Gaussian widths are drawn from $\mathcal{N}(900, 50^2)$. The temporal dynamics of backgrounds are simulated from a constrained Gaussian random walk process with steps drawn from $\mathcal{N}(0, 2^2)$, then clipped to be non-negative and Gaussian smoothed temporally with a variance of 60 frames. We also simulate motion of the field of view as 2-D translations. The translational shift in each direction is simulated from a constrained Gaussian random walk process with steps drawn from $\mathcal{N}(-0.2d, 1)$, where $d$ is the current amount of shift. Lastly, we add a $\mathcal{N}(0, 0.1^2)$ Gaussian noise to the entire simulated data. The activity of neurons is multiplied by a scalar before combining with the background activity and noise. We call this scalar 'signal level.'

To validate the accuracy of Minian output, we simulate data with different signal level and number of cells. The signal levels we use are 0.2, 0.4, 0.6, 1.0, 1.4, and 1.8. The number of cells we use is 100, 300, and 500. On the other hand, to benchmark the performance of Minian, we simulate data with different number of frames and cells. The number of frames varies from 4000 to 28,000 with a step size of 8000. The number of cells we use is 100, 300, and 500.

## Matching neurons for validation

To compute different metrics of the accuracy of Minian output, we first need to match the putative neurons from Minian output with neurons from ground truth. To obtain this mapping, we first compute the max projection of spatial footprints across all neurons. We then register the max projection of putative spatial footprints to the max projection of ground-truth spatial footprints by estimating a translational shift between the two max projection images. After correcting for translational shifts, we compute the center of mass for all neurons, from which we obtain an N × M pairwise distance matrix, where N and M are number of neurons detected by Minian and number of ground-truth neurons, respectively. We then calculate an optimal mapping by solving the linear assignment problem of minimizing the total cost (distance) of a particular cell mapping. Lastly, we threshold the resulting mapping by discarding any matched cell that has a distance larger than 15 pixels.

## Classification of place cells

We use the spatially binned averaged 'firing' rate calculated from spike signals to classify whether each cell is a place cell. A place cell must simultaneously satisfy three criteria: spatial information criterion, stability criterion, and place field size criterion. To determine whether a cell has significant spatial information or stability, we obtain a null distribution of the measurements (spatial information and stability) with a bootstrap strategy, where we roll the timing of activity by a random amount for each cell 1000 times. The observed spatial information or stability is defined as significant if it exceeds the 95th percentile of its null distribution (p<0.05). For the spatial information criterion, we use the joint information between 'firing' rate and an animal's location measured in bits per 'spike.' For the stability criterion, we calculate the Fisher's z-transformation of the Pearson correlation coefficient between spatial 'firing' patterns across different trials within a recording session. A trial is defined as the time that the animal runs from one end of the linear track to the other and returns to the starting location. We calculate the z-transformed correlation between the odd number of trials and the even number of trials, as well as between the first half of the trials and the second half of the trials. We then average these two measures of correlations and use that as the measure of stability for a cell. Lastly, for the place field size criterion, we define the place field of each cell as the longest contiguous spatial bin where the averaged 'firing' rate exceeded the 95th percentile of all averaged firing rate bins. A cell must have a place field larger than 4 cm (i.e., two spatial bins) to pass the place field size criterion.

## Animals

Adult male C57/BL6J mice from Jackson Laboratories were used for all testing. Animals were housed in a temperature, humidity, and light-controlled vivarium down the hall from the experimental testing rooms with lights on at 7 am and off at 7 pm. Water was restricted to maintain a body weight of 85–90%. Water deprivation consisted of allotting the animal ~1 ml of water per day, including water obtained during testing. Water not obtained during testing was given after the testing period. Animals were acclimated to handling for 5–7 days prior to training/testing. All experiments were

performed in accordance with relevant guidelines and regulations approved by the Institutional Animal Care and Use Committee of Icahn School of Medicine at Mount Sinai (reference # IACUC-2017-0361, protocol # 17-1994).

## Tested hardware specifications

The hardware specifications of computers that have effectively run Minian are summarized in the following table.

**Appendix 1—table 1.** A list of computers tested with Minian with specifications (listed roughly by increasing computation power).

| Manufacture | Model | CPU | RAM | Storage | Operating system |
|---|---|---|---|---|---|
| Custom-built | Carbon | AMD Ryzen Threadripper 2950 × 4.4 GHz × 16 | 128 GB | 2TB SSD | Ubuntu 18.04 |
| Microsoft | Surface Pro 6 | Intel Core i5-8250U 1.6 GHz × 4 | 8 GB | 256GB SSD | Windows 10 |
| Dell | Precision 5,530 | Intel Core i5-8400H 2.5 GHz × 4 | 16 GB | 256GB SSD | Ubuntu 18.04 |
| Apple | MacBook Pro 152 | Intel Core i7-8559U 2.7 GHz × 4 | 16 GB | 1TB SSD | macOS 10.14 Mojave |
| Custom-built | Amethyst | Intel Xeon E5-1650 3.6 GHz × 6 | 128 GB | 6TB HDD | Ubuntu 17.1 |

## List of dependencies

**Appendix 1—table 2.** A list of open-source packages and the specific versions on which Minian depends.

| Package | Version |
|---|---|
| av | 7.0 |
| Bokeh | 1.4 |
| Bottleneck | 1.3 |
| cairo | 1.16 |
| CVXPY | 1.0 |
| Dask | 2.11 |
| Datashader | 0.1 |
| distributed | 2.11 |
| ecos | 2.0 |
| FFmpeg | 4.1 |
| FFTW | 3.3 |
| HoloViews | 1.12 |
| IPython | 7.12 |
| ipywidgets | 7.5 |
| Jupyter | 1.0 |
| Matplotlib | 3.1 |
| natsort | 7.0 |
| netCDF4 | 1.5 |
| NetworkX | 2.4 |
| Node.js | 13.9 |
| Numba | 0.48 |
| NumPy | 1.18 |

*Appendix 1—table 2 Continued on next page*

*Appendix 1—table 2 Continued*

| Package | Version |
| --- | --- |
| openCV | 4.2 |
| pandas | 1.0 |
| Panel | 0.8 |
| Papermill | 2.0 |
| param | 1.9 |
| pip | 20.0 |
| pyFFTW | 0.12 |
| Python | 3.8 |
| SciPy | 1.4 |
| scs | 2.1 |
| statsmodels | 0.11 |
| tifffile | 2020.2 |
| tqdm | 4.43 |
| xarray | 0.15 |
| Zarr | 2.4 |
| MedPy | 0.4 |
| SimpleITK | 1.2 |

## Comparison of algorithms in related pipelines

**Appendix 1—table 3.** List of algorithm implementations in different pipelines.
For a lot of steps, different algorithm implementation can be chosen by the user based on features of the data. In such cases, we only list the default and most commonly used algorithms here.

| Step | Minian implementation | CalmAn implementation | MIN1PIPE implementation | Critical parameters |
| --- | --- | --- | --- | --- |
| Denoising | Median filter | None | Anisotropic filter | Spatial window size of the filter |
| Background removal | Morphological top-hat transform | None | Morphological top-hat transform | Spatial window size of the top-hat transform |
| Motion correction | FFT-based translational motion correction | Nonrigid patch-wise translational motion correction (NoRMCorre) | Mix of translational motion correction and Demons diffeomorphic motion correction | Different |
| Initialization | Seed-based with peak-noise ratio and KS-test refinement | Pixel-wise correlation and peak-noise ratio thresholding | Seed-based with GMM, peak-noise ratio and KS-test refinement | Threshold for correlation and peak-noise ratio |
| Spatial and temporal updates | CNMF with CVXPY as deconvolution backend | CNMF-E with Oasis as deconvolution backend | CNMF with CVX MATLAB package as deconvolution backend | Noise cutoff frequency Expected size of neurons Sparse penalty |

GMM: Gaussian mixture model; KS: Kolmogorov–Smirnov; CNMF: constrained non-negative matrix factorization.

