## [Editor Report]

An increasing number of systems neuroscience experiments involve imaging neural activity using head-mounted epifluorescent microscopes, or "miniscopes". A growing community has been using the Minian software package to process the imaging data into a useful form for subsequent analysis. The Minian team has done an excellent job of exposing the various parameters involved to be easily accessible to users while maintaining a performant robust tool. This work presents Minian and is in many ways an exemplar of how open source software can be presented in journal form.

---

## [Decision Letter]

**Decision letter after peer review:**

Thank you for submitting your work entitled "Minian: An Open-source Miniscope Analysis Pipeline" for further consideration by *eLife*. Your revised article has been reviewed by three peer reviewers, including Caleb Kemere as the Reviewing Editor and Reviewer #1, and the evaluation has been overseen by a Laura Colgin as the Senior Editor. The reviewers have opted to remain anonymous.

The reviewers were unanimous in concluding that there was considerable value in presenting Minian to the field. In particular they appreciated (as you have described) the insight into parameter selection that the Minian codebase allows. However, there was concern that the manuscript does not adequately compare Minian with existing tools (e.g., CaImAn) in terms of batch run time (i.e., after the parameters have been selected and one wants to churn through data) and final performance/accuracy (i.e., are the ROIs "as good"?). Given that memory needs are a claimed advantage, a comparison here would also be valuable. Reviewer 3 has used simulated data for the ROI comparison and includes code in their review.

*Reviewer #1 (Recommendations for the authors):*

The advent of open source miniscopes has significantly expanded the population of experimenters who are excited to record activity-dependent fluorescence in freely moving animals. Unfortunately, they often find that once they have solved the challenges that arise in acquiring the data, a new challenge awaits – extracting the neural activity for analysis. While there are existing open source tools for extracting activity-dependent fluorescence traces from video data, these packages are not particularly accessible for investigators with significant experience in computational code. For novices, they can be impenetrable. The Minian toolbox aims to solve the problem of data analysis while also bringing along novices, illuminating different parameter choices, and making each step of the process more understandable. We found that, to a great extent, it achieves this aim, and thus we think that it rises to the level of significant interest to a broad audience. We did have some concerns with the current manuscript. In particular, limited evidence is provided to allow comparison of results, performance and efficiency between Minian and other established packages (e.g., how in terms of processing time, or in terms of the results).

If Minian's major contribution is usability and accessibility (lines 134-135) it is not clear why a new package had to be created rather than extending CaImAn or MIN1PIPE. The former does have out-of-core support, where memory usage can be reduced by limiting the number of parallel processes. A table detailing the methods/algorithms used by each of these packages may be useful for comparison purposes.

Lines 113-115: The text indicates that data may still be split into patches of pixels, so it is unclear how merging based on overlapping parts can be avoided. Aren't the overlaps necessary because putative neurons may span patch boundaries? Perhaps the authors could also compare performance of temporal splitting and non-splitting data processing. Will temporal splitting lead to comparable performance?

Lines 117-119. Had to look at documentation website to figure out how to limit memory usage--put this in paper. Include some graphs to show the tradeoff between memory usage and computational time.

Please comment on motion correction when there is a rotation in the FOV?

For motion correction, provide additional information about how the template is generated (e.g. how many frames used) and any user-defined parameters.

Lines 205-206: Will the same mask be applied to all frames before motion correction? Or the mask will be applied to this certain frame and then be applied to all frames after motion correction?

Line 499: write the function used for optimization.

Paragraph starting at line 576 would benefit from some equations for clarity's sake.

Implement manual merging as in original CNMF-E paper?

Line 699: define preprocessed with respect to Figure 1--this is post motion-correction presumably?

Line 725, For Figure 14, what is the fps for the data taken? Only frame number was shown in the right side column, if the fps is 20, then 150 frames correspond to about 10 sec of calcium transient, which is on the long side for a healthy neuron.

Line 826-828: Is dimension here referring to x and y dimensions of the frame? Please elaborate how each slice on one dimension can be processed independently.

Discuss robustness to long data (where CaImAn seems to fail). Using Minian, was CNMF performed on each msCam video individually? If so, what happens when a calcium event occurs near the video boundaries?

Line 879: What would be an example number that you would consider as "significantly high (see below) stability"? It is not clear what "see below" was referring to in terms of high stability. Example provided for place field size criterion only.

*Reviewer #2 (Recommendations for the authors):*

Dong et al. presents "Minian" (Miniscope Analysis), a novel, open-source analysis pipeline for single photon in vivo calcium imaging studies. Minian aims to allow for users with relatively little computational experience to extract cell location and corresponding calcium traces in order to process and analyze neural activity from imaging experiments. It does so by enhancing the visualization of changes to free parameters during each step of data processing and analysis. Importantly, Minian operates with a relatively small memory demand, making it feasible for use on a laptop. In this manuscript, the authors validate Minian's ability to reliably extract calcium events across different brain regions and from different cell types. The manuscript is well-organized, clear, and easy to follow. Importantly, this manuscript describes a resource which complements previous open-sourced tools built by the authors (e.g. the UCLA Miniscope), as well as other, similar systems. Collectively, these tools comprise an affordable, open-source, easy to use system for single photon calcium imaging and analysis in freely-behaving mice. This project will impact the field by providing open-source, user-friendly, customizable, intuitive software for the analysis of single photon in vivo calcium imaging experiments across a variety of imaging systems.

1) Page 5, line 100. In regards to visualization of parameter changes and resulting outcomes using Minian, the authors state that "This feature provides users with knowledge about which parameter values should be used to achieve the best outcomes and whether the results appear accurate." What do the authors mean by "best outcomes" and "appear accurate"? Is there a way the authors can suggest to rigorously test for data accuracy rather than looking to see if data "appear accurate"? Do the authors plan to provide any guidance for standard parameters/variables to use for brain regions and/or cell types? These issues are somewhat highlighted in the limitations section at the end, but I think there could be some more effort put into providing more objectivity to this approach.

2) Page 9, line 240. As part of their "denoising" step, the authors pass their frames through a median filter. A number of filters can be used for denoising (e.g. Gaussian). The authors should justify why this filter is used and potentially compare it to other filters.

3) Page 9 describes setting the window size to avoid "salt and pepper" noise on the one end, and "over-smoothing" on the other. Minian allows for the visualization of distinct settings of the window size, however, it is left up to the user to select the "best" window. This allows for user bias in selection of an optimal window. One thing that would be useful is if the window size selected could be compared to the expected cell diameter, which could then be reported alongside final results.

4) Page 23, line 671. For registering cells across multiple recording sessions, the authors "discard all matches that result in conflict". Are users notified of number of discarded matches and is there a way to discard one match (eg cell A to B) but not another (Cell B to C)?

5) Page 26, like 750. The authors use Minian to process and analyze calcium imaging data and ezTrack to extract animal location. The two are time-locked using timestamps and further analyses are performed to correlate imaging and behavioral data. Do the authors synchronize their imaging and behavioral data using additional software? If so, can they provide information onto how best to synchronize and analyze such data?

*Reviewer #3 (Recommendations for the authors):*

Context:

Brain imaging techniques using miniaturized microscopes have enabled deep brain imaging in freely behaving animals. However, very large background fluctuations and high spatial overlaps intrinsic to this recording modality complicate the goal of efficiently extracting accurate estimates of single-neuronal activity. The CNMF-E algorithm uses a refined background model to address this challenge and has been implemented in the CaImAn analysis pipeline. It yields accurate results, but is demanding both in terms of computing and memory requirements. Later an online adaptation of the CNMF-E algorithm, OnACID-E, has been presented which dramatically reduces its memory and computation requirements.

Strengths:

Here the authors use a simpler background model that has been suggested by the authors of the MIN1PIPE pipeline. After the background has been (approximately) removed the data can be processed using the standard CNMF algorithm. The latter requires some user specified parameters and the interactive visualizations provided by Minian greatly simplify the setting of these parameters. Thus Minian's learning curve is less steep than that of other pipelines. Using a less refined background model and libraries that support parallel and out-of-core computation, Minian requires less memory than the batch CNMF-E algorithm.

Weakness:

While Minian's memory requirements are low compared to the batch CNMF-E algorithm, its online adaptation OnACID-E is even less demanding. The authors base their insights mainly on visual inspection, and fail to thoroughly validate the accuracy of their extracted temporal traces using simulated ground truth data as well as various real datasets as was done in the original CNMF-E (as well as the MIN1PIPE) paper. Own analysis of simulated ground truth data indicates that Minian uses less memory but also performs less well than CaImAn's CNMF-E implementation with regard to computational processing time and accuracy of extracted deconvolved calcium traces.

Upshot:

Minian is a user-friendly single-photon calcium imaging analysis pipeline. It's interactive visualizations facilitate accessibility and provide an easy means to set key parameters. In practice, Minian's slower computational processing time compared to other pipelines is likely offset by less user time spent on parameter specification. I expect other pipelines will take up the idea put forward by Minian to use interactive visualizations for the latter.

You primarily focus on interactive visualizations. I had no trouble installing Minian and thoroughly enjoyed the visualizations afforded by xarray, dask, bokeh and holoviews.

Basing your pipeline on established algorithms makes one hopeful to get accurate results, but since you are not just providing a GUI for a well validated pipeline but taking inspiration from CaImAn, MIN1PIPE and your own previous work, that's not guaranteed. In light of this, I don't find the Results section convincing.

Showing only averaged results and comparing to a null distribution leaves the reader with the impression that the pipeline performs better than random, which is not that impressive. Your pipeline actually does quite well and deserves better promotion.

You need to rework the Results section and follow along the lines of the CNMF-E and MIN1PIPE paper:

* compare to simulated ground truth data

- false positives/false negatives

- most importantly, how accurate are the extracted deconvolved traces that would be used in the downstream analysis?

* compare to multiple real datasets

- false positives/ false negatives using manual annotation as (imperfect) 'ground truth'

- show individual traces

Because one of your sale pitches is Minian's low memory demand, you need to include a figure illustrating the scaling of processing time and memory with dataset size (FOV, #frames, #neurons), cf. Figure 8 of the CaImAn paper.

A good starting point would be the simulated data of the CNMF-E paper. I used that data to gain some experience with Minian as a potential future user, and to compare it to CaImAn, with which I already have some familiarity.

After parameter tweaking both yielded a perfect F1 score of 1. However, the denoised/deconvolved traces obtained with CaImAn had an average correlation of 0.9970/0.9896 with ground truth, whereas Minian only yielded average correlations of 0.9669/0.9304.

The processing time on my Desktop/MacBookPro was 46.6s/169.1s for CaImAn and 140.9s/359.8s for Minian (using 12/4 workers, 2 threads). The peak memory was 31.698GB/4.508GB for CaImAn and 11.382GB/5.534GB for Minian.

Thus, at least in my hands, CaImAn outperforms Minian, presumably due to the more refined background model, and I'll likely stick with the former. Though I grant you that parameter selection is easier in Minian. I wished there was a pipeline with the best of both.

[Editors' note: further revisions were suggested prior to acceptance, as described below.]

Thank you for resubmitting your work entitled "Minian An Open-source Miniscope Analysis Pipeline" for further consideration by *eLife*. Your revised article has been evaluated by Laura Colgin (Senior Editor) and a Reviewing Editor.

The manuscript has been improved but there are some remaining issues that need to be addressed, as outlined below:

Please address the comments of the reviewers below. In particular, the issue of reproducibility (i.e., missing data source files) is important, as is the concern of reproducing Figure 18.

*Reviewer #1 (Recommendations for the authors):*

Most concerns were addressed, but a couple of clarifications are still needed. First, with regard to boundary conditions for chunks, based on our understanding of your response, the text should be updated to clarify that neither spatial nor temporal chunking can be used for the core CNMF computation (as it computes across both space and time). How does this limitation constrain performance?

Second, with regard to chunking, there is a long description of motion correction. It seems that the initial chunk has a minimum size of 3, which will be aligned, then 9 frames will be aligned, and so on in powers of three (27, 243, etc). Is this correct? Perhaps some numerical example would be helpful for this text description.

*Reviewer #2 (Recommendations for the authors):*

The revisions adequately addressed all of my concerns.

*Reviewer #3 (Recommendations for the authors):*

The authors did a good job addressing the vast majority of my concerns and improving the paper.

My main concern is lacking reproducibility, contrary to the claims in the "Transparent Reporting File":

While the repo for figure 1-14 and figure 20 can be found at

https://github.com/denisecailab/Minian_paper_data, the data source files that are supposed to be hosted as zip files on a google drive are missing.

The repo for figure 15-19 https://github.com/denisecailab/Minian-validation does not exist, or is not public.

I have not been able to reproduce Figure 18.

I repeated (6x) and concatenated the 2000frames of your demo pipeline, and processed them using 4 parallel processes. (The only parameters in Caiman's demo pipeline adjusted were gSig=(5,5); gSiz=(21,21); min_corr=.75; min_pnr=7; yielding ~280 cells). The results should thus be in the same ballpark as Figure 18, 300 cells, 12k frames.

However, the runtime was similarly about 25 min whether using Minian or Caiman, and whether running on a Linux Desktop or MacBookPro.

The reported memory seems about right for my Desktop, however, my MacBookPro processed the data using Caiman and merely 10GB of RAM, the same amount of memory as Minian.

Your demo pipeline doesn't run on my MacBookPro (with M1 chip), it fails in update_background step, and I had to increase the memory limit for each worker from 2 to 2.5GB.

I have the following remaining issues:

line 542: You need to subtract the neural activity AC before projecting, f = b'(Y-AC).

line 635: Is the 'input movie data' the data after subtracting the contributions of all the other neurons and background bf? Otherwise projecting the input data on the cell doesn't properly demix them if other components overlap. The least squares solution of min_c ||y-Ac|| is (A'A)^{-1}A'y, not A'y.

Figure 15: Why not put error bars for the correlations like you do in Figure 16? If the error bars are smaller than the symbols, say so in the paper. (E.g. standard error of the median via bootstrap)

Figure 16: Why does the correlation saturate at ~0.8? Shouldn't it approach ->1 for noise->0?

line 851: Is the data described in a previous publication you missed to cite? Frames, FOV, rate?

line 895: "despite the fact…". Why despite? This is expected behavior when the number of parallel processes is set to be constant, without putting a limit on memory.

line 933: Is the data described in [35]? I.e. 320x320@20Hz.

---

## [Author Response]

Reviewer #1 (Recommendations for the authors):If Minian's major contribution is usability and accessibility (lines 134-135) it is not clear why a new package had to be created rather than extending CaImAn or MIN1PIPE. The former does have out-of-core support, where memory usage can be reduced by limiting the number of parallel processes. A table detailing the methods/algorithms used by each of these packages may be useful for comparison purposes.

While our initial intention was to standardize input/output of CaImAn and visualize how changing some of the parameters affected the output of the pipeline, we realized an increasing need to re-implement functions to improve efficiency and integration with visualizations which evolved into a separate pipeline, Minian. With that said, Minian is greatly inspired by CaImAn and MIN1PIPE as we highlighted in the manuscript.

As the reviewer noted, the memory usage of CaImAn is related to the number of parallel processes and limiting this number will limit the performance gain from multi-processor CPUs. Moreover, the memory usage by each process still scales linearly with recording length. In contrast, due to the flexible chunking implementation of Minian, the memory usage can be limited to as low as 2GB per process and stays constant across recording length. We have included a detailed comparison of memory usage between the two pipelines in the Results section. We have also added a table in supplemental information detailing the different algorithm implementation across pipelines (Table 3).

The new comparison of memory usage starts at page 32 line 895 in the manuscript:

“As shown in Figure 18, the run time of both Minian and CaImAn scales linearly as a function of input recording length, with Minian consistently achieving shorter running times. On the other hand, the peak memory usage of CaImAn scales linearly with recording length, despite the fact that the number of parallel processes was set to be constant. At the same time, the peak memory usage of Minian stays mostly constant across increasing number of frames. This is likely due to the flexible chunking implementation of Minian (See Parallel and out-of-core computation with dask), where Minian was able to break down computations into chunks in both the spatial and the temporal dimensions depending on which way is more efficient. In contrast, CaImAn only splits data into different spatial chunks (patches), resulting in a linear scaling of memory usage with recording length for each chunk-wise computation.”

Lines 113-115: The text indicates that data may still be split into patches of pixels, so it is unclear how merging based on overlapping parts can be avoided. Aren't the overlaps necessary because putative neurons may span patch boundaries? Perhaps the authors could also compare performance of temporal splitting and non-splitting data processing. Will temporal splitting lead to comparable performance?

We have updated the text regarding the chunking algorithm to make it clear that both spatial and temporal splitting happens in the pipeline. In order to avoid the boundary problem posed by the reviewer, each step in the pipeline performs either spatial or temporal splitting independently. For example, if a smoothing filter is applied to each frame of the data, then temporal splitting is performed. In contrast, if pairwise pixel correlation is calculated, then spatial splitting is performed.

The relevant updated text starts at page 37 line 1012 in the manuscript:

“In Minian, we use a modern parallel computing library called dask to implement parallel and out-of-core computation. Dask divides the data into small chunks along all dimensions, then flexibly merges the data along some dimensions in each step. We leverage the fact that each step in our pipeline can be carried out chunk by chunk independently along either the temporal (frame) dimension or the spatial (height and width) dimensions, thus requiring no interpolation or special handling of borders when merged together, producing results as if no chunking had been done. For example, motion correction and most pre-processing steps that involve frame-wise filtering can be carried out on independent temporal chunks, whereas the spatial update step in CNMF algorithm and computation of pixel correlations can be carried out on independent spatial chunks. Consequently, our pipeline fully supports out-of-core computation, and memory demand is dramatically reduced. In practice, a modern laptop can easily handle the analysis of a full experiment with a typical recording length of up to 20 minutes. Dask also enables us to carry out lazy evaluation of many steps where the computation is postponed until the result is needed, for example, when a plot of the result is requested. This enables selective evaluation of operations only on the subset of data that will become part of the visualization and thus helps users to quickly explore a large space of parameters without committing to the full operation each time.”

Lines 117-119. Had to look at documentation website to figure out how to limit memory usage--put this in paper. Include some graphs to show the tradeoff between memory usage and computational time.

In order to further clarify how to use and implement Minian, we have now included a “Setting up” section in the main text. This includes the process of configuring the computation cluster, including the number of parallel processes and memory limit. We have also included a new graph of tradeoff between memory usage and computation time as the reviewer suggested (Figure 19).

The new “Setting up” section starts at page 5 line 187 in the manuscript:

“The first section in the pipeline includes house-keeping scripts to import packages and functions, defining parameters, and setting up parallel computation and visualization. Most notably, the distributed cluster that carries out all computations in Minian are set up in this section. By default, the cluster runs locally with multi-core CPUs, however it can be easily scaled up to run on distributed computers. The computation in Minian is optimized such that in most cases the memory demand for each process/core can be as low as 2GB. This allows the total memory usage of Minian to roughly scale linearly with the number of parallel processes. The number of parallel processes and memory usage of Minian are completely limited and managed by the cluster configuration allowing users to easily change them to suit their needs.”

The discussion on computation tradeoff starts at page 32 line 896 in the manuscript:

“Additionally, we run Minian and CaImAn with different number of parallel processes on the simulated dataset with 28000 frames and 500 cells. As expected, with more parallel processes the performance improves and the run time decreases but at the same time the total peak memory usage increases. The tradeoff between run time and peak memory usage are shown in Figure 19. In conclusion, these results show that in practice, Minian is able to perform as fast as CaImAn, while maintaining near constant memory usage regardless of input data size.”

Please comment on motion correction when there is a rotation in the FOV?

In practice, we have found that the motion of the brain is mostly translational, which is corrected by the motion correction algorithms currently implemented. We have updated the text describing motion correction and made it explicit that currently no rotational correction is implemented.

The updated description for motion correction starts at page 12 line 324 in the manuscript:

“We use a standard template-matching algorithm based on cross-correlation to estimate and correct for translational shifts. […] After the estimation of shifts, the shift in each direction is plotted across time and visualization of the data before and after motion correction is displayed in Minian (see Figure ??, top right).”

For motion correction, provide additional information about how the template is generated (e.g. how many frames used) and any user-defined parameters.

We have updated the text describing the divide-and-conquer implementation of motion correction, as well as how templates are generated for each chunk and how they are used to estimate motion (described in the above updated motion correction section).

Lines 205-206: Will the same mask be applied to all frames before motion correction? Or the mask will be applied to this certain frame and then be applied to all frames after motion correction?

The visualization provided by Minian only records an arbitrary mask drawn by the user, and it’s up to the user to decide when and how to use that mask to apply to a subset of the data. With that, the most common usage of this function (which is included in the default pipeline) is to use the mask to sample a part of the FOV and use the smaller FOV for estimation of motion only. The motion correction will then be carried out on the full data without masking.

Line 499: write the function used for optimization.

We have now included a formal definition of the optimization problem in the main text.

Paragraph starting at line 576 would benefit from some equations for clarity's sake.

We have now included a formal definition of the optimization problem in the main text.

Implement manual merging as in original CNMF-E paper?

As suggested, we have detailed an additional visualization at the end of the CNMF steps allowing for manual merging. The new visualization shows the spatial footprints and temporal activities of cells alongside the raw data. This helps the user easily determine the quality of the output cells, and the user can choose to discard or merge cells within the visualization tool.

The description of the new visualization and manual curation tool starts at page 24 line 710 in the manuscript:

“Minian provides an interactive visualization to help the users manually inspect the quality of putative cells and potentially merge or drop cells. […] In this way, only the new label is saved and no data is modified, allowing the user to repeat or correct the manual curation process if needed.”

Line 699: define preprocessed with respect to Figure 1--this is post motion-correction presumably?

Yes, the preprocessed video refer to the data after motion-correction. We have now included clarification in the main text.

Line 725, For Figure 14, what is the fps for the data taken? Only frame number was shown in the right side column, if the fps is 20, then 150 frames correspond to about 10 sec of calcium transient, which is on the long side for a healthy neuron.

The data was acquired with 30 fps and we have now clarified that in figure caption.

Line 826-828: Is dimension here referring to x and y dimensions of the frame? Please elaborate how each slice on one dimension can be processed independently.

We have now updated the section describing the flexible chunking implementation and out-of-core computation support, to clarify that the dimension to be sliced would be chosen for each step to make sure chunks on that dimension can be processed independently (this is included in the updated description of the chunking algorithm above).

Discuss robustness to long data (where CaImAn seems to fail). Using Minian, was CNMF performed on each msCam video individually? If so, what happens when a calcium event occurs near the video boundaries?

In Minian, the individual msCam videos are concatenated from the beginning. However, due to the flexible chunking implementation in Minian, we can choose the dimension to split up for each step. As a result, we can avoid boundary problems suggested by the reviewer. For example, we can split in the spatial dimensions when temporal updates are performed which leads to dramatically reduced memory demand and robustness to longer recordings. We have included a new benchmark figure in the Results section demonstrating this (Figure 18).

Line 879: What would be an example number that you would consider as "significantly high (see below) stability"? It is not clear what "see below" was referring to in terms of high stability. Example provided for place field size criterion only.

We have updated the section describing the place cell criteria moving the shuffling process description to the beginning of the paragraph, making it clear that “significantly high” was defined as 95th percentile of the null distribution of the specific metric.

The updated description starts at page 39 line 1095 in the manuscript:

“We use the spatially-binned averaged ‘firing’ rate calculated from spike signals to classify whether each cell is a place cell. A place cell must simultaneously satisfy three criteria: a spatial information criterion, a stability criterion, and a place field size criterion. To determine whether a cell has significant spatial information or stability, we obtain a null distribution of the measurements (spatial information and stability) with a bootstrap strategy, where we roll the timing of activity by a random amount for each cell 1000 times. The observed spatial information or stability is defined as significant if it exceeds the 95th percentile of its null distribution (p < 0.05).”

Reviewer #2 (Recommendations for the authors):1) Page 5, line 100. In regards to visualization of parameter changes and resulting outcomes using Minian, the authors state that "This feature provides users with knowledge about which parameter values should be used to achieve the best outcomes and whether the results appear accurate." What do the authors mean by "best outcomes" and "appear accurate"?

Indeed, the language could be misleading as it is hard to objectively define “best outcomes”. We have updated the manuscript to make it clear that it is up to the user to determine the outcome that fits best with their expectation.

Is there a way the authors can suggest to rigorously test for data accuracy rather than looking to see if data "appear accurate"?

In the revised manuscript we have included extensive validation of Minian with simulated datasets, along with comparisons with the existing pipeline, CaImAn. The accuracy remains high and indistinguishable from CaImAn as long as proper parameters are chosen. This suggests that Minian can perform accurately.

The relevant text from the new validation section starts at page 27 line 803 in the manuscript:

"To compare the results objectively, we first matched the resulting putative cells from the output of Minian or CaImAn to the simulated ground truth (See Matching neurons for validation for details). [...] This is likely due to noise and background contaminating the true signal. Overall, these results show that the Minian and CaImAn pipelines perform similarly well in terms of output accuracy on simulated datasets.”

Do the authors plan to provide any guidance for standard parameters/variables to use for brain regions and/or cell types?

Minian provides a pipeline with default parameters that works best on the datasets we have tested, which consist mainly of single-photon hippocampal CA1 recordings. Additionally, we provide detailed explanation and visualization for each step which equips the users with knowledge to explore and adjust the parameters for datasets obtained across different brain regions/cell types. However, since there are many variables that determine the features of the data and the optimal parameters, it is hard to provide a pre-determined set of parameters that works universally on certain brain regions/cell types.

These issues are somewhat highlighted in the limitations section at the end, but I think there could be some more effort put into providing more objectivity to this approach.

In order to provide objectivity to the Minian pipeline, we have included an updated validation section, which includes results from both simulated data and real experimental data. The outputs of Minian are validated with respect to ground truth and existing pipelines. Our results from both simulated and real datasets suggest that Minian performs as well as CaImAn.

In addition to the validation with simulated data presented above, we also included validation on real datasets with respect to existing pipeline CaImAn, which starts at page 30 line 854 in the manuscript:

“We next validated Minian with experimental datasets. The data was collected from hippocampal CA1 regions in animals performing a spatial navigation task. 6 animals with different density of cells were included in the validation dataset. […] Overall, these results suggest that the output of Minian is highly similar to CaImAn when analyzing experimental datasets.”

2) Page 9, line 240. As part of their "denoising" step, the authors pass their frames through a median filter. A number of filters can be used for denoising (e.g. Gaussian). The authors should justify why this filter is used and potentially compare it to other filters.

While Minian implements other filters for denoising purpose, including Gaussian, bilateral, and anisotropic filters seen in MIN1PIPE, we found the median filter works best for the “salt and pepper” type of noise introduced by CMOS sensors. Hence we choose median filter as the default for the pipeline and leave the other filters available as options.

3) Page 9 describes setting the window size to avoid "salt and pepper" noise on the one end, and "over-smoothing" on the other. Minian allows for the visualization of distinct settings of the window size, however, it is left up to the user to select the "best" window. This allows for user bias in selection of an optimal window. One thing that would be useful is if the window size selected could be compared to the expected cell diameter, which could then be reported alongside final results.

While Minian allows the user to choose the “best” window, it is indeed more helpful to suggest optimal parameters based on features of the data whenever possible. We have now included the suggestion to set the window size of the median filter to the expected cell diameter.

4) Page 23, line 671. For registering cells across multiple recording sessions, the authors "discard all matches that result in conflict". Are users notified of number of discarded matches and is there a way to discard one match (eg cell A to B) but not another (Cell B to C)?

Currently there is no way to discard one match over the other, since there is no clear intuitive way to handle the conflicts and allowing such manual correction would likely induce user bias. That said, users can implement other matching criteria to best fit their requirements.

5) Page 26, like 750. The authors use Minian to process and analyze calcium imaging data and ezTrack to extract animal location. The two are time-locked using timestamps and further analyses are performed to correlate imaging and behavioral data. Do the authors synchronize their imaging and behavioral data using additional software? If so, can they provide information onto how best to synchronize and analyze such data?

The open-source Miniscope data acquisition software synchronizes the start and end time of the calcium video and behavioral video. The time stamps from each frame collected from the Miniscope and behavioral cameras are then synched so each calcium imaging frame corresponds to the correct behavioral video frame.

Reviewer #3 (Recommendations for the authors):You primarily focus on interactive visualizations. I had no trouble installing Minian and thoroughly enjoyed the visualizations afforded by xarray, dask, bokeh and holoviews.Basing your pipeline on established algorithms makes one hopeful to get accurate results, but since you are not just providing a GUI for a well validated pipeline but taking inspiration from CaImAn, MIN1PIPE and your own previous work, that's not guaranteed. In light of this, I don't find the Results section convincing.Showing only averaged results and comparing to a null distribution leaves the reader with the impression that the pipeline performs better than random, which is not that impressive. Your pipeline actually does quite well and deserves better promotion.You need to rework the Results section and follow along the lines of the CNMF-E and MIN1PIPE paper:* compare to simulated ground truth data- false positives/false negatives- most importantly, how accurate are the extracted deconvolved traces that would be used in the downstream analysis?* compare to multiple real datasets- false positives/ false negatives using manual annotation as (imperfect) 'ground truth'- show individual traces

We thank the reviewer for their thoughtful and detailed comments. We have updated our manuscript to include validation with both simulated and real datasets detailed in the Results section.

The validation with simulated data starts at page 27 line 794 in the manuscript:

“We first validated Minian with simulated datasets. We synthesized different datasets with varying number of cells and signal levels based on existing works. […] This is likely due to noise and background contaminating the true signal. Overall, these results show that the Minian and CaImAn pipelines perform similarly well in terms of output accuracy on simulated datasets.”

In particular, the validation of extracted deconvolved traces starts at page 28 line 828 in the manuscript:

“Additionally, we want to validate the deconvolved signal from Minian output, since this is usually the most important output for downstream analysis. […] Overall, these results suggest that Minian can produce deconvolved signals that are faithful to ground truth and suitable for downstream analysis.”

The validation with real data starts at page 30 line 854 in the manuscript:

“We next validated Minian with experimental datasets. The data was collected from hippocampal CA1 regions in animals performing a spatial navigation task. 6 animals with different density of cells were included in the validation dataset. […] Overall, these results suggest that the output of Minian is highly similar to CaImAn when analyzing experimental datasets.”

Additionally, we have performed validation on real datasets with manual annotations as the reviewer suggested. We collected manual annotations from two independent labelers who were instructed to find circular-shaped neurons with a characteristic calcium decay dynamic. The labelers were provided with a minimally pre-processed video, with the bright vignetting artifact and motion of the brain corrected. The labelers used Fiji (ImageJ) to visualize the pre-processed video and the free-hand drawing tool to produce ROIs. This annotation procedure provides a binary mask for each putative neuron. To generate consensus ground truth, we match the binary ROIs across the two labelers using the same procedure used for matching neurons for validation, and consider only the matched ROIs as ground truth neurons. To obtain the spatial footprints for each ground truth neuron, we take the mean projection across the binary masks from the two labelers. Hence the spatial footprints generated in this way can only take values 0, 0.5 or 1. To obtain the temporal dynamic for each ground truth neuron, we project the pre-processed video onto the spatial footprints by calculating a mean activity of each pixel weighted by each spatial footprints.

The result is summarized in Author response image 1. Our results indicate that both pipelines capture most of the consensus footprints. However, inconsistencies exist across the two pipelines, and even across human labelers, shown by a relatively low F1 score among the two human labelers. We recognize that this is probably due to the complex nature of single-photon recordings, and in a lot of cases it is hard to determine whether an ROI is a real cell even for experienced human labelers. Furthermore, since we are benchmarking with two different pipelines, we are unable to generate imperfect ground truth by seeding the CNMF algorithm with manual labels, since the choice of CNMF implementation will likely bias the resulting correlations. Instead, we have to use more naive method to generate ‘ground truth’, namely taking the mean of manual labeled spatial footprints and projecting the raw data onto spatial footprints to use as temporal ground truth. However, such approach has significant drawback: First, the ‘ground truth’ spatial footprints can only take 3 discrete values, and their correlation with real-valued spatial footprints from analysis pipelines would be hard to interpret. Secondly, the ‘ground truth’ temporal activity contains noise and is contaminated with cross-talk from nearby cells. A perfect correlation with such ‘ground truth’ would actually indicate that the analysis pipeline failed to perform denoise and demixing of calcium activities. Due to these drawbacks, we decided to only present these results as response to reviewer comments but did not include them in the main text, as we believe these results might be hard to interpret and would be potentially misleading to general readers.

**Author response image 1. sa2fig1:** Validation of Minian with experimental datasets. An example field of view from one of the experimental datasets is shown in panel A, with contours of detected ROIs overlaid on top. ROIs labeled by human labelers are shown in green, where solid lines indicate ROIs labeled by both labelers (labeled as “Manual-Consensus”) and dashed lines indicate ROIs labeled by one of the labelers (labeled as “Manual-Mismatch”). The F1 scores, spatial footprints correlation and temporal dynamics correlation are plotted for the two pipelines in panel B. The F1 scores of the two human labelers are also included in the plot for comparison (labeled as “Manual”). The F1 scores, spatial footprints correlation and temporal dynamics correlation were all not significantly different across the two pipelines (One-way ANOVA, p > 0.05).

Because one of your sale pitches is Minian's low memory demand, you need to include a figure illustrating the scaling of processing time and memory with dataset size (FOV, #frames, #neurons), cf. Figure 8 of the CaImAn paper.A good starting point would be the simulated data of the CNMF-E paper. I used that data to gain some experience with Minian as a potential future user, and to compare it to CaImAn, with which I already have some familiarity.After parameter tweaking both yielded a perfect F1 score of 1. However, the denoised/deconvolved traces obtained with CaImAn had an average correlation of 0.9970/0.9896 with ground truth, whereas Minian only yielded average correlations of 0.9669/0.9304.The processing time on my Desktop/MacBookPro was 46.6s/169.1s for CaImAn and 140.9s/359.8s for Minian (using 12/4 workers, 2 threads). The peak memory was 31.698GB/4.508GB for CaImAn and 11.382GB/5.534GB for Minian.Thus, at least in my hands, CaImAn outperforms Minian, presumably due to the more refined background model, and I'll likely stick with the former. Though I grant you that parameter selection is easier in Minian. I wished there was a pipeline with the best of both.

We followed the suggested path for benchmarking Minian performance and have include a new benchmarking figure in the Results section. We found that processing time for both Minian and CaImAn scales linearly as a function of recording length, but Minian has a near constant peak memory usage across different recording length, while peak memory usage for CaImAn scales linearly to recording length.

The relevant text in the new benchmarking section starts at page 32 line 885 in the manuscript:

“To see how the performance of Minian scales with different input data size, we synthesized datasets with varying number of cells and number of frames (recording length). […]This allows the users to process much longer recordings with limited RAM resources.”

[Editors' note: further revisions were suggested prior to acceptance, as described below.]

The manuscript has been improved but there are some remaining issues that need to be addressed, as outlined below:Please address the comments of the reviewers below. In particular, the issue of reproducibility (i.e., missing data source files) is important, as is the concern of reproducing Figure 18.

We thank the reviewer’s for their in-depth comments and validation. We addressed the issue of reproducibility in this revision. Specifically, we have used google drive as the data storage service in our previous submission, and we were not aware that it requires a google account to access even when we made it public. Hence, we have now switched to a dedicated data sharing service Figshare and made sure the code (github repos) as well as raw data are publicly available without credentials. We have also included convenient scripts on github to automatically download all the necessary data and produce the correct folder structure, making our results more reproducible. Additionally, we were also able to reproduce Reviewer 3’s results, and we addressed them in the specific responses below. In response to reviewer’s comments, we have included additional parameters/conditions in our benchmarking and validation experiments with simulated data (i.e., Figure 15, 16, 18, 19).

Reviewer #1 (Recommendations for the authors):Most concerns were addressed, but a couple of clarifications are still needed. First, with regard to boundary conditions for chunks, based on our understanding of your response, the text should be updated to clarify that neither spatial nor temporal chunking can be used for the core CNMF computation (as it computes across both space and time). How does this limitation constrain performance?

We apologize for the confusion. Although the core CNMF involves computations across both space and time, the spatial and temporal update steps are carried out as separate steps, and either spatial or temporal chunking can be independently applied to each step. We have updated the description in the text to clarify this point (starting on line 1033):

“For example, motion correction and most pre-processing steps that involve frame-wise filtering can be carried out on independent temporal chunks, whereas computation of pixel correlations can be carried out on independent spatial chunks. Similarly, during the core CNMF computation steps, spatial chunking can be used during update of spatial footprints, since spatial update is carried out pixel by pixel. Meanwhile, temporal chunking can be used when projecting the input data onto spatial footprints of cells, which is usually the most memory-demanding step. Although the optimization step during the temporal update is computed across all frames and no temporal chunking can be used, we can still chunk across cells, and in practice the memory demand in this step is much smaller comparing to other steps involving raw input data. Consequently, our pipeline fully supports out-of-core computation, and memory demand is dramatically reduced.”

Second, with regard to chunking, there is a long description of motion correction. It seems that the initial chunk has a minimum size of 3, which will be aligned, then 9 frames will be aligned, and so on in powers of three (27, 243, etc). Is this correct? Perhaps some numerical example would be helpful for this text description.

This is correct and we thank the reviewer for the suggestion. We have updated the relevant description to include a numerical example (starting on line 337):

“After the registration, the 3 chunks that have been registered are treated as a new single chunk and we again take the max projection to use as a template for further registration. In this way, the number of frames registered in each chunk keeps increasing in powers of three (3, 9, 27, 81 etc.), and we repeat this process recursively until all the frames are covered in a single chunk and the whole movie is registered. Since the motion correction is usually carried out after background removal, we essentially use cellular activity as landmarks for registration.”

Reviewer #3 (Recommendations for the authors):The authors did a good job addressing the vast majority of my concerns and improving the paper.My main concern is lacking reproducibility, contrary to the claims in the "Transparent Reporting File":While the repo for figure 1-14 and figure 20 can be found athttps://github.com/denisecailab/Minian_paper_data, the data source files that are supposed to be hosted as zip files on a google drive are missing.The repo for figure 15-19 https://github.com/denisecailab/Minian-validation does not exist, or is not public.

We apologize for this error. Along with this resubmission, we have made sure both repos are available publicly, and all the raw data can be downloaded online without any login credentials.

I have not been able to reproduce Figure 18.I repeated (6x) and concatenated the 2000frames of your demo pipeline, and processed them using 4 parallel processes. (The only parameters in Caiman's demo pipeline adjusted were gSig=(5,5); gSiz=(21,21); min_corr=.75; min_pnr=7; yielding ~280 cells). The results should thus be in the same ballpark as Figure 18, 300 cells, 12k frames.However, the runtime was similarly about 25 min whether using Minian or Caiman, and whether running on a Linux Desktop or MacBookPro.

We thank the reviewer for the careful validation of our results. Upon further investigation, we realized that visualization, especially the generation of two validation videos, takes a significant amount of processing time in Minian pipeline. These steps are enabled by default in the demo pipeline, but was not included in our benchmark in previous submission. Once we include the visualization steps we were able to reproduce the running time similar to reviewer’s results (~25min for 12K frames for both pipelines). We have now updated Figure 18 to include results with and without visualizations for both pipelines (There is only one major visualization in CaImAn hence the results are very similar within CaImAn). The relevant parts in the main text are updated as follows (starting on line 907):

“As shown in Figure *18*, the run time of both Minian and CaImAn scales linearly as a function of input recording length. The exact running times vary depending on number of cells as well as whether visualization is included in the processing, but in general the running time is similar across both pipelines.”

The reported memory seems about right for my Desktop, however, my MacBookPro processed the data using Caiman and merely 10GB of RAM, the same amount of memory as Minian.

Indeed, we are able to reproduce the low memory footprint of CaImAn on a MacBookPro as well. We believe this is due to the fact that OS X system uses all available space on the machine’s boot partition as swap/virtual memory. Hence, if we only observe usage of physical memory, it appears CaImAn (or any program) would only use as little as the amount of free physical memory, while in fact swap memory would be used to account for any additional memory demand. We have modified our benchmarking code to track the usage of swap memory as well and plot them together with physical memory usage as in Author response image 2. Our results suggest that even in OS X, although the physical memory usage is similar across pipelines, the swap memory usage for CaImAn scales linearly as the size of input data increases, while the swap memory usage of Minian stays constantly close to 0. Although this result implies that a user can potentially run analysis on very large datasets on a MacBookPro without Minian, the user still needs special configuration of the operating system if analysis is not performed on a Mac hardware. In general, regardless how an operating system handles memory allocation, Minian still has a much smaller memory demand comparing to existing pipelines.

**Author response image 2. sa2fig2:** Benchmarking of computational performance in OS X. Data with varying number of cells and frames were processed through Minian and CaImAn without visualization. The run time (top) and peak physical memory usage (middle) and peak swap memory usage (bottom) were recorded and plotted as a function of frame number.

Your demo pipeline doesn't run on my MacBookPro (with M1 chip), it fails in update_background step, and I had to increase the memory limit for each worker from 2 to 2.5GB.

While we don’t have a MacBookPro with M1 chip to test, we expect that sometimes the algorithms would require more than 2GB of memory per worker to finish depending on the exact hardware/software configuration, and it is common for Minian users to get around the memory error by increasing the memory limit a bit. We have updated the “Setting up” section in the main text to include this information (starting on line 191):

“The computation in Minian is optimized such that in most cases the memory demand for each process/core can be as low as 2GB. However, in some cases depending on the hardware, the state of operating system and data locality, Minian might need more than 2GB per process to run. If a memory error (KilledWorker) is encountered, it is common for users to increase the memory limit of the distributed cluster to get around the error. Regardless of the exact memory limit per process, the total memory usage of Minian roughly scales linearly with the number of parallel processes.”

I have the following remaining issues:line 542: You need to subtract the neural activity AC before projecting, f = b'(Y-AC).

We have updated the texts with correct information (starting on line 546):

“After the spatial footprint of the background term is updated, we subtract the neural activity (AC) from the input data to get residual background fluctuations. Then the temporal activity of background term is calculated as the projection of residual onto the new background spatial footprint, where the raw activities of each pixel is weighted by the spatial footprint.”

line 635: Is the 'input movie data' the data after subtracting the contributions of all the other neurons and background bf? Otherwise projecting the input data on the cell doesn't properly demix them if other components overlap. The least squares solution of min_c ||y-Ac|| is (A'A)^{-1}A'y, not A'y.

The reviewer is correct and we did account for the activity from overlapping cells and subtract them from the projection. We agree that in theory we should use the least square solution as the target activity. However, in practice we use an empirical projection following the block-coordinate decent update rule to include sparsity regularization and achieve better computational performance. This implementation can demix activities from overlapping cells and is consistent with CaImAn. The exact equation is described as Eq. 6 in (Friedrich, Giovannucci, and Pnevmatikakis 2021). We have updated the description (starting on line 598):

“First, we subtract the background term from the input data, leaving only the noisy signal from cells. We then project the data onto the spatial footprints of cells, obtaining the temporal activity for each cell. Next we estimate a contribution of temporal activity from neighboring overlapping cells using the spatial footprints of cells, and subtract it from the temporal activity of each cell.”

Figure 15: Why not put error bars for the correlations like you do in Figure 16? If the error bars are smaller than the symbols, say so in the paper. (E.g. standard error of the median via bootstrap)

For Figure 15 we did not generate multiple datasets for the same condition (signal level and number of cells). The reason is that we don’t expect much variance in any of the metric for the same conditions.

Figure 16: Why does the correlation saturate at ~0.8? Shouldn't it approach ->1 for noise->0?

We have fine-tuned some parameters and included additional data points with higher signal level. The results now indicate that the correlation asymptotes and approaches 1 when signal level is higher than 1. We have updated the text and figures (starting on line 842):

“The resulting correlation is summarized in Figure *16* A. Our results indicate that the deconvolved output from Minian is highly similar to ground truth spikes when signal level is high, and the correlation asymptote and approach 1 when signal level is higher than 1. The lower correlation corresponding to low signal level is likely due to the background and noise contamination being stronger than signal.”

line 851: Is the data described in a previous publication you missed to cite? Frames, FOV, rate?

This data has not been previously published. We apologize for the missing details. We have now added description of the data (starting on line 864):

“The data was collected from hippocampal CA1 regions in animals performing a spatial navigation task. 6 animals with different density of cells were included in the validation dataset. The recordings are collected with 608 x 608 pixels at 30 fps and lasts 20 min (~36000 frames).”

line 895: "despite the fact…". Why despite? This is expected behavior when the number of parallel processes is set to be constant, without putting a limit on memory.

We agree it can be confusing regarding the expected memory usage. We have now reworded this sentence as the following (starting on line 910):

“On the other hand, the peak memory usage of CaImAn scales linearly with recording length when the number of parallel processes was set to be constant.”

line 933: Is the data described in [35]? I.e. 320x320@20Hz.

We apologize for the missing information. We have now added description of the data (starting on line 946):

“Briefly, animals were trained to run back and forth on a 2 m linear track while wearing a Miniscope to obtain water rewards available at either end. The time gap between each session was 2 days. We record calcium activity in dorsal CA1 region with a FOV of 480 x 752 pixels collected at 30 fps. Each recording session lasts 15 min (~27000 frames).”